# SOCRATIC MODELS: COMPOSING ZERO-SHOT MULTIMODAL REASONING WITH LANGUAGE

**Andy Zeng, Maria Attarian, Brian Ichter, Krzysztof Choromanski, Adrian Wong,**
**Stefan Welker, Federico Tombari, Aveek Purohit, Michael S. Ryoo, Vikas Sindhwani,**
**Johnny Lee, Vincent Vanhoucke, Pete Florence**

Google

## ABSTRACT

We investigate how multimodal prompt engineering can use language as the intermediate representation to combine complementary knowledge from different pretrained (potentially multimodal) language models for a variety of tasks. This approach is both distinct from and complementary to the dominant paradigm of joint multimodal training. It also recalls a traditional systems-building view as in classical NLP pipelines, but with prompting large pretrained multimodal models. We refer to these as Socratic Models (SMs): a modular class of systems in which multiple pretrained models may be composed zero-shot via multimodal-informed prompting to capture new multimodal capabilities, without additional finetuning. We show that these systems provide competitive state-of-the-art performance for zero-shot image captioning and video-to-text retrieval, and also enable new applications such as (i) answering free-form questions about egocentric video, (ii) engaging in multimodal assistive dialogue with people (e.g., for cooking recipes), and (iii) robot perception and planning. We hope this work provides (a) results for stronger zero-shot baseline performance with analysis also highlighting their limitations, (b) new perspectives for building multimodal systems powered by large pretrained models, and (c) practical application advantages in certain regimes limited by data scarcity, training compute, or model access.

## 1 INTRODUCTION

Large language models (LLMs) (Chowdhery et al., 2022; Devlin et al., 2018; Brown et al., 2020; Thoppilan et al., 2022; Chen et al., 2021) are capable of performing complex language tasks by conditioning (i.e., "prompting") the model with several input examples (few-shot) or instructions describing the task (zero-shot). Prompting methods such as "Chain of Thought" (Wei et al., 2022; Kojima et al., 2022) and subsequent work have shown to be particularly effective for a wide range of reasoning benchmarks, and shed light on new opportunities to quickly re-purpose large pretrained models for new tasks without additional data collection or finetuning. Given the empirical success of prompt engineering for language-based tasks, and given the rise of language models grounded on other modalities (e.g., visual-language models, VLMs, such as CLIP (Radford et al., 2021; Li et al., 2021a; Wang et al., 2021; Jain et al., 2021)), we investigate: to what extent can prompt engineering be extended to perform multimodal reasoning between such models?

We study how *language as the intermediate representation* can be used to compose large pretrained models and address a variety of multimodal reasoning problems. Specifically, the premise is that different pretrained (potentially multimodal) language models contain distinct knowledge: VLMs are trained on image captions, while LLMs are additionally trained on other data (spreadsheets, fictional novels, and standardized test questions, etc.), but they can be combined together using language via prompt engineering to build new application-specific programs, without further model finetuning. Central to this approach is *multimodal prompt engineering*, which may include e.g., in-context substitutions of visual entities from a VLM into the input prompt of an LLM, or listing candidate output text predictions from an LLM and re-ranking their relevance to images or videos using a VLM. The prompts can be designed either manually (Schick & Schütze, 2020; Reynolds & McDonell, 2021) or automatically (Gao et al., 2020; Shin et al., 2020), and offer a distinct yet compatible option with the predominant paradigm of jointly training unified multimodal models (Hu & Singh, 2021) on big data (Jia et al., 2021). While there exists some prior work in this area (Yang et al., 2021), this paper aims to provide a more comprehensive view of the capabilities of systems built in this way, discuss both their advantages and disadvantages in relation to modern and classical multimodal paradigms, and present additional analysis on how to evaluate such systems.

Extensive experiments with vision, language, and audio modalities show that on various problems, multimodal prompt engineered systems can be quantitatively competitive with zero-shot state-of-the-art on standard benchmarks including (i) image captioning on MS COCO, (ii) contextual image captioning and description (improving 11.3 (Kreiss et al., 2021) to 38.8 captioning CIDEr on Concadia), and (iii) video-to-text retrieval (from 40.3 (Portillo-Quintero et al., 2021) to 44.7 zero-shot R@1 on MSR-VTT (Xu et al., 2016)). The approach also gives rise to new opportunities to address classically challenging problems in one domain, by reformulating it as a problem in another – for example, formulating video understanding as a reading comprehension problem (Rajpurkar et al., 2018), for which modern LLMs are proficient (Sec. 5.1). This enables baselines for new applications such as (i) open-ended reasoning for egocentric perception (Fig. 4), (ii) multimodal assistive dialogue to

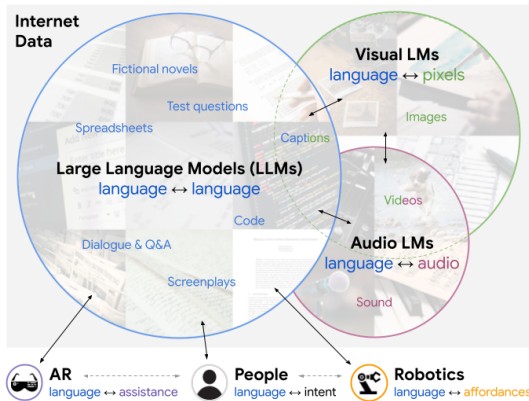

Fig. 1: Large pretrained models across different domains learn complementary forms of knowledge, and language is an intermediate representation by which these models can exchange information to generate joint predictions for new multimodal tasks, without finetuning. Multimodal prompting these models can enable new applications in data-scarce domains e.g., augmented reality (AR), human robot interaction (HRI).

guide a user through a cooking recipe, and (iii) robot perception-driven planning for sequential pick and place. Multimodal prompt engineering can be viewed as a systems approach that re-visits classic NLP pipelines (Manning et al., 2014) but with a modern twist – large pretrained (Bommasani et al., 2021) models as the modules, multimodal domains as the problem setting. Natural language as middleware exhibits the benefits of compositional generality (Hupkes et al., 2020), yields practical benefits in domains where data is scarce, but also presents clear limitations on expressing more fine-grained detailed information between modalities. We discuss these and directions for future work. Open-source code is available at https://socraticmodels.github.io.

## 2 RELATED WORK

We are interested in enabling a variety of multimodal (Ngiam et al., 2011) applications by prompt engineering large pretrained models. This can be viewed as a form of transfer learning (Caruana, 1997; Thrun, 1998), where knowledge from *pretraining tasks* (e.g., text completion, image-text similarity) is applied to new *downstream tasks* (e.g., image captioning, robot planning). We accordingly review related paradigms in pretraining, multimodal models, pipelined systems, and prompting.

**Pretraining weights** is a dominant paradigm for transfer learning with deep models, in which model weights from pretraining tasks are used to initialize a subset of model parameters for the target task, which are either (a) left frozen, or (b) finetuned. Pretraining deep models has been studied extensively in the unsupervised setting (Hinton et al., 2006; Bengio et al., 2006; Vincent et al., 2008; Raina et al., 2007; Mesnil et al., 2012), in the supervised setting was perhaps popularized by ImageNet (Deng et al., 2009) pretraining (Girshick et al., 2014; Donahue et al., 2014; Zeiler & Fergus, 2014; Sermanet et al., 2013), and has been ubiquitous in NLP (Mikolov et al., 2013; Pennington et al., 2014; Dai & Le, 2015; Ramachandran et al., 2016; Peters et al., 2018; Devlin et al., 2018; Brown et al., 2020). Downstream target tasks may require additional domain-specific model architectures or training procedures. In multimodal training, it is common to leave sub-portions of models e.g., weights associated with one but not other modalities, frozen for downstream tasks (Zhai et al., 2021; Florence et al., 2019; Tsimpoukelli et al., 2021; Zakka et al., 2022).

**End-to-end joint training of multiple modalities** is a common approach to multimodal learning (Tsimpoukelli et al., 2021; Lu et al., 2019; Mokady et al., 2021; Gao et al., 2021; Song et al., 2022a; Zellers et al., 2022). For each task $i$ one may obtain a large multimodal dataset and train a task-specific map $f^i_{\theta_i}$ with parameters $\theta_i$, some of which may come from pretrained weights, either frozen or finetuned. A benefit of this approach is that it follows the recipe of: (1) curate a big dataset, (2) train a big model, which given enough data and compute can be formidable (Sutskever et al., 2014). Combining weights from large pretrained models with multimodal joint training, several works have achieved strong results for a number of downstream multimodal applications

including VLMs with LLMs for image captioning (e.g., CLIP with GPT-2) (Mokady et al., 2021), video understanding (e.g., CLIP with BERT (Gao et al., 2021)), visual Q&A e.g., (Song et al., 2022a) and audio-language models (ALMs) and LLMs for speech and text modeling e.g., (Song et al., 2022b; Bapna et al., 2022). These systems are often finetuned on task-specific data, and while this paradigm is likely to be preferred in data-rich domains, our results suggest that SMs can be a strong alternative for data-scarce, training-compute-limited, or model-access-limited applications.

**Pipelined or probabilistic multimodal systems** can be considered a broad category of alternatives to end-to-end joint training, which was popular before the emergence of large multimodal end-to-end-trained systems. One primary example for such systems comes from classical NLP pipelines (Manning et al., 2014; Tenney et al., 2019), in which engineers lay out a sequence of application-specific steps, such as parts-of-speech tagging and named entity recognition. A similar class of systems may involve modularizing the problem not by a sequence of steps, but rather by probabilities from different modalities: e.g., a Bayesian approach where one model is used as a prior and the other as evidence – with which models from different modalities may perform joint inference (Karpagavalli & Chandra, 2016; Ahn et al., 2022). One prominent example is in automatic speech recognition: different language models can be trained separately, then transfer knowledge to a speech-to-text system via priors (Karpagavalli & Chandra, 2016). The notion of "Mixture-of-Experts" ((Jordan & Jacobs, 1994), see (Masoudnia & Ebrahimpour, 2014) for a review) is also common for combining the outputs of multiple models, including multimodal (Liu et al., 2019a).

**Zero-shot and few-shot prompting** recently have been shown to be highly effective for transfer learning (Brown et al., 2020; Xie et al., 2021; Min et al., 2022). In this approach, large pretrained language models are zero-shot or few-shot *prompted*, as in asked to provide a certain type of response, without training, to perform a new task. Further methods such as chain-of-thought prompting (Wei et al., 2022; Kojima et al., 2022) have shown that even simple prompting modifications can have a profound impact on target task performance (Wei et al., 2022; Chowdhery et al., 2022; Kojima et al., 2022) and enable new capabilities. Likewise, creative prompting has been shown to be essential to generating high-quality text-to-image results (Oppenlaender, 2022). Our work builds on these works, by extending prompt engineering methods to address multimodal reasoning problems.

## 3    Multimodal Prompt Engineering

We study a class of systems in which multiple large pretrained models may be composed with prompt engineering to perform new multimodal tasks that each model otherwise would struggle to do independently. We refer to these systems as "Socratic Models" (SM) – loosely inspired by the Socratic Method, since models with different commonsense knowledge can work together to arrive at a conclusion. These systems employ *multimodal prompt engineering*, which may encompass: (i) designing input text prompts to elicit specific responses from LLMs, (ii) ranking the relevance of text predictions against other modalities (e.g., pixels with VLMs), (iii) using text predictions to call subprograms, or (iv) combining text outputs from multiple modalities by substituting parts of an input prompt to an LLM (via in-context substitution). This can be viewed as re-examining a systems approach similar to classical NLP pipelines (Manning et al., 2014; Tenney et al., 2019), but for multimodal domains, and directly using natural language as the intermediate representation by which the modules exchange information. This is driven by the compositional generality of language (Hupkes et al., 2020; Keysers et al., 2019), and can be applied to domains in which data is scarce, models are only available through APIs without source-code access, or it may be prohibitively expensive to train a new large multimodal model. SMs are both distinct from, and may be complementary to, models that are jointly multimodally trained (Sec. 2).

These systems are perhaps most intuitively understood through examples, which are provided in Sec. 4 and 5, but a definition is as follows. A task-specific SM system $f_{\text{SM}}$ : inputs $\rightarrow$ outputs may be described as a computation graph, with nodes as a set of modules $\{f^i_{\mathcal{M}^i}\}$, and the edges of the graph represent inter-module communication through language. Each $\mathcal{M}$ is some (multimodal) model or external API, and each module $f$ assists in transforming the output of one $f$ into a form of language that a connected $f'$ may use for further inference. For visualization, outputs from LLMs are blue, VLMs green, ALMs (audio-language) purple, prompt text gray, user inputs orange, VLM-chosen LLM outputs green-underlined blue, and ALM-chosen LLM outputs purple-underlined blue. A key component of our approach is in-context substitution, where information from a non-language domain is substituted into a language prompt, used as input to an LLM for contextual reasoning.

One specific way is to variable-substitute text descriptions of entities from other modalities into the prompt. An example of this is shown in an activity-recognition example in Fig. 2: activity = $f_{\text{LLM}}(f_{\text{VLM}}(f_{\text{LLM}}(f_{\text{ALM}}(f_{\text{LLM}}(f_{\text{VLM}}(\text{video}))))))$, in which (i) the VLM detects visual entities, (ii) the LLM suggests sounds that may be heard, (iii) the ALM chooses the most likely sound, (iv) the LLM suggests possible activities, (v) the VLM ranks the most likely activity, (vi) the LLM generates a summary. Some form of such multimodal prompting with in-context substitution is central to all of our demonstrated SM examples (Sec. 4 and 5). Note that this example involves multiple back-and-forth interactions, including calling the same model multiple times, forming "closed-loop" feedback between models.

Informally the graph can be interpreted as composing pretrained models to "talk to each other", but in practice certain models may need pre- or post-processing to produce language. For example, image-text similarity VLMs, e.g., CLIP (Radford et al., 2021), do not inherently produce text, but can be made to perform zero-shot detection from a large pre-existing library of class category names, and return the top-$k$ matching categories. Accordingly, although our example SM systems require no training, the interactions between models are scripted with prompt templates. This presents practical benefits in certain settings: new applications can be creatively programmed, without data or training, and with only API-level-access to models required.

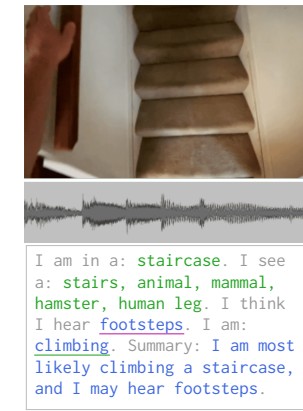

I am in a: `staircase`. I see a: `stairs, animal, mammal, hamster, human leg`. I think I hear `footsteps`. I am: `climbing`. Summary: `I am most likely climbing a staircase, and I may hear footsteps.`

Fig. 2: Multimodal prompt engineered systems can zero-shot annotate egocentric images with a summary of the person's activities. Information from multiple modalities (language, audio) can denoise predictions from any one specific modality (vision).

## 4 EXPERIMENTS: METHODS AND RESULTS

The goal of this section is to both provide example systems (see code for implementations) and also evaluate performance relative to both state-of-the-art jointly-trained multimodal systems, and prior multimodal-engineered systems. We quantitatively evaluate example systems on: image captioning (Sec. 4.1), contextual image captioning (Sec. 4.2), and video-to-text retrieval (Sec. 4.3).

### 4.1 IMAGE CAPTIONING ON MS COCO CAPTIONS: VLM + LLM

I am an intelligent image captioning bot. This image is a {img_type}. There {num_people}. I think this photo was taken at a {place1}, {place2}, or {place3}. I think there might be a {object1}, {object2}, {object3},... in this {img_type}. A creative short caption I can generate to describe this image is:

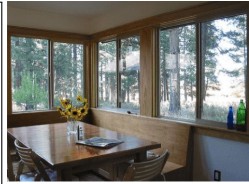

**SM (ours):** This image shows an inviting dining space with plenty of natural light.

**ClipCap:** A wooden table sitting in front of a window.

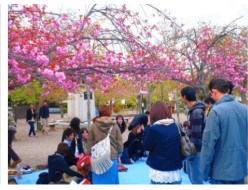

**SM (ours):** People gather under a blossoming cherry tree, enjoying the beauty of nature together.

**ClipCap:** Students enjoying the cherry blossoms.

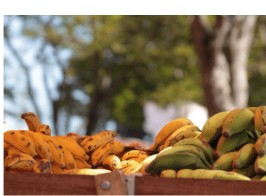

**SM (ours):** At the outdoor market, you can find everything from plantains to Japanese bananas.

**ClipCap:** A bunch of bananas sitting on top of a table.

Fig. 3: VLM with LLM prompting (left) can zero-shot generate captions for Internet images (e.g., from MS COCO), and can be as expressive as task-specific finetuned methods such as ClipCap (Mokady et al., 2021).

**Method.** We can generate image captions via multimodal prompt engineering between a VLM and LLM – i.e., via caption = $f_{\text{VLM}}^3(f_{\text{LLM}}^2(f_{\text{VLM}}^1(\text{image})))$. First (1), the VLM is used to zero-shot detect different place categories (Places356 (Zhou et al., 2016)), object categories (from Tencent ML-Images (Wu et al., 2019)), image type ({*photo, cartoon, sketch, painting*}) and the number of people {*no people, one person, ..., several people*}. The top-$k$ ranked in each category can then be substituted into an LLM prompt as context, shown in Fig. 3, left. Second (2), given the VLM-informed language prompt, a causal LLM (i.e., for text completion) generates several $n$ candidate captions. For this step, we use a non-zero next-token sampling temperature (e.g., 0.9 for GPT-3), to return sufficiently diverse, but reasonable results across the $n$ candidates. Finally (3), these $n$ captions are then ranked by the VLM with the image, and the highest scoring caption is returned.

**Results.** Tab. 1 shows comparisons with recent state-of-the-art methods on MS COCO Captions dataset (Chen et al., 2015; Lin et al., 2014). We evaluate over a random sampled subset of 100 images from the test split (Karpathy & Fei-Fei, 2015), so that GPT-3 API runtime costs are more affordable for reproducibility (∼$150 USD per run with $n = 20$ ranked candidate captions per image). Metrics from baselines on this subset are a close estimate of the full test set metrics (shown in Appendix). Our system outperforms the zero-shot state-of-the-art ZeroCap (Tewel et al., 2021) with a CIDEr (Vedantam et al., 2015) score 18.0 → 50.1, but does not perform as well as methods such as ClipCap (Mokady et al., 2021) which are directly finetuned on the training set. Our method tends to generate verbose captions (see qualitative examples in Fig. 3), but may naturally score lower on captioning metrics if they do not match the dataset's distribution of caption labels. This performance gap narrows if the LLM is additionally few-shot prompted with 3 random captions from the training set, bringing CIDEr scores up to 76.3, exceeding the performance of MAGIC (Su et al., 2022) which finetunes the text generator on the training set's unpaired captions. We hope future work may use these results as a stronger zero-shot baseline, while also relieving its limitations e.g., CLIP-based ranking VLMs would struggle to express detail-rich image captions.

| Method | BLEU-4 | METEOR | CIDEr | SPICE | ROUGE-L |
|---|---|---|---|---|---|
| *ClipCap (Mokady et al., 2021) | 40.7 | 30.4 | 152.4 | 25.2 | 60.9 |
| †MAGIC (Su et al., 2022) | 11.4 | 16.4 | 56.2 | 11.3 | 39.0 |
| ZeroCap (Tewel et al., 2021) | 0.0 | 8.8 | 18.0 | 5.6 | 18.3 |
| SMs 0-shot (ours) | 10.0 | 16.2 | 50.1 | 10.8 | 36.1 |
| SMs 3-shot (ours) | **18.2** | **20.5** | **76.3** | **13.9** | **43.7** |

\* finetuned on full training set with image-text pairs.
† finetuned on unpaired training set, zero-shot on image-text pairs.

Tab. 1: Comparisons suggest SMs perform well on zero-shot image captioning over a subset of MS COCO test examples.

**Ablations.** We also run captioning experiments both with (i) changing the wording of the LLM prompts (Kojima et al., 2022; Liu et al., 2021), and (ii) ablations that remove VLM categories. We observe (Tab. 2) that changing the prompt wording from "a likely, short caption" to "a caption" or to "a likely, creative caption" decreases performance – the captions generated by these alternatives tend to be overly verbose, and

| Prompt Changes | BLEU-4 | METEOR | CIDEr | SPICE | ROUGE-L |
|---|---|---|---|---|---|
| SMs 0-shot (original) | 10.0 | 16.2 | 50.1 | 10.8 | 36.1 |
| *Wording Changes* | | | | | |
| remove "likely, short" | 5.5 | 16.7 | 35.3 | 11.0 | 31.3 |
| replace "short" w/ "creative" | 5.5 | 16.3 | 32.3 | 10.2 | 30.2 |
| remove "intelligent" | 10.7 | 16.6 | 56.4 | 10.5 | 36.5 |
| *Categories Ablations* | | | | | |
| remove object categories | 4.3 | 11.3 | 25.9 | 6.5 | 29.0 |
| remove place categories | 8.0 | 16.8 | 47.1 | 10.3 | 36.3 |
| remove number of people | 8.9 | 17.2 | 48.7 | 11.6 | 36.6 |

Tab. 2: Prompt wording changes and category ablations.

are less likely to match the distribution of (short) captions in the COCO dataset. Surprisingly, removing "intelligent" from the description "intelligent image captioning bot" seems to slightly improve performance. We also observe that removing entities from the VLM (objects, places, number of people) also tends to reduce performance, most substantially when object categories are removed.

## 4.2 Contextual Image Description on Concadia: VLM + LLM

**Method.** We also demonstrate a *contextual* captioning system, using a similar method to the previous section (Sec. 4.1) but with in-context substituting article text (below), comprising $f_{LLM}^2(f_{VLM}^1(\text{image}), \text{context})$, for which we find good results without requiring VLM re-ranking.

```
I am an intelligent image captioning bot. The article is about: "{article_text}". In this image, I think I see
a {object1}, {object2}, {object3},... A short caption for this image is:
```

**Results.** Concadia (Kreiss et al., 2021) is a dataset for generating contextual image captions and descriptions, conditioned on an image and associated article text. In particular, image descriptions describe visual content in the image (e.g., "portrait of a man with a beard in a suit") used for accessibility, while captions link images to article text (e.g., "photo of Abraham Lincoln"). We evaluate on the full Concadia test split with 9,691 images (shown in Tab. 3). Our system performs favorably over the

| Method | Caption Generation | Description Generation |
|---|---|---|
| Kreiss et al. (2021) | 11.3 | 17.4 |
| SMs 0-shot (ours) | 38.8 | 23.0 |
| SMs 3-shot (ours) | **59.6** | **27.3** |
| SMs w/ description | 93.8 | – |

Tab. 3: On generating contextual image captions and descriptions (CIDEr) from Concadia, SMs zero-shot outperform task-specific methods e.g., (Kreiss et al., 2021) that finetune on the training set.

prior best method, (Kreiss et al., 2021), which finetunes on the training set of 77,534 images; with a CIDEr score improvement 11.3 → 38.8 for generated image captions, and 17.4 → 23.0 for generated

image descriptions (59.6 and 27.3 respectively for 3-shot). We also report numbers for generating captions conditioned on the image, article text, and *ground truth* description. This achieves a CIDEr score of 93.8 and suggests an upper bound of performance if SMs are used with VLMs that can produce accurate image descriptions (additional observations in the Appendix). Overall, these results suggest SM-based systems can be used to generate descriptive texts that improve content accessibility for the low vision community. Further, from a system-building perspective, this experiment shows that leveraging stronger LLMs is promising, even if models are not jointly trained: the prior method (Kreiss et al., 2021) jointly trained a multimodal system with LLM pretrained weights, but our system uses a considerably more powerful LLM (GPT-3 "davinci") model without requiring joint training, which would be infeasible due to the unavailability of GPT-3's source code.

## 4.3 VIDEO-TO-TEXT RETRIEVAL ON MSR-VTT: VLM + LLM + ALM

**Method.** We also address video-to-text retrieval, a common video understanding task, by using both audio and visual data. We improve on a prior approach (Portillo-Quintero et al., 2021) which computes a CLIP-based video-and-text similarity measure for one-to-many nearest neighbor matching. Adding in audio information, our system transcribes audio with speech-to-text ALMs (Bapna et al., 2022) for automatic speech recognition (ASR e.g., via Google Cloud speech-to-text API (gcl)), then summarizes the transcripts with an LLM using the following prompt:

```
I am an intelligent video captioning bot.' I hear a person saying: "{transcript}". Q: What's a short video
caption for this video? A: In this video,
```

We compute similarity scores of the generated summary to the set of captions with a masked LLM (e.g., with sentence similarity from RoBERTa (Liu et al., 2019b)), and use those scores to re-weight the CLIP-based ranking from Portillo-Quintero et al. (2021). For videos with sufficiently-long transcripts ($\geq 145$ characters), the matching score is: $(CLIP \text{ (caption)} \cdot CLIP \text{ (video}')) \times (RoBERTa \text{ (caption)} \cdot RoBERTa \text{ (}GPT\text{-}3(prompt, Speech2Text \text{ (audio}'))))$, where $\cdot$ represents normalized dot product of embeddings, and $\times$ represents scalar multiplication. If there is no audio or the transcript is too short, we default to Portillo-Quintero et al. i.e., the dot product of CLIP text embeddings and averaged CLIP image embeddings of all video frames $CLIP\text{(caption)} \cdot CLIP\text{(video}')$.

**Results.** We evaluate on MSR-VTT (Xu et al., 2016), noted in other recent works (Gao et al., 2021; Cheng et al., 2021) as a popular benchmark for video-to-text retrieval. We compare our method with zero-shot methods, as well as finetuned methods specifically trained on MSR-VTT. Results show that our method outperforms zero-shot state-of-the-art (Tab.4). Since our system uses Portillo-Quintero et al. (2021) to process CLIP features but additionally incorporates LLM reasoning on speech-to-text transcripts,

|  |  | MSR-VTT Full | | | | |
|---|---|---|---|---|---|---|
| Category | Method | R@1↑ | R@5↑ | R@10↑ | MdR↓ | Audio |
| *Finetuned* | JEMC (Mithun et al., 2018) | 12.5 | 32.1 | 42.4 | 16.0 | yes |
| | Collab. Experts (Liu et al., 2019a) | 15.6 | 40.9 | 55.2 | 8.3 | yes |
| | CLIP2Video (Fang et al., 2021) | **54.6** | **82.1** | **90.8** | **1.0** | no |
| *Zero-shot* | Portillo-Quintero et al. (2021) | 40.3 | 69.7 | 79.2 | **2.0** | no |
| | SMs (ours) | **44.7** | **71.2** | **80.0** | 2.0 | yes |

Tab. 4: Video-to-text retrieval results on MSR-VTT (Xu et al., 2016) dataset with the original 'full' test set. Differentiated here are methods which train on the MSR-VTT dataset (*finetuning*), compared with *zero-shot* methods, which do not. Also noted: whether the methods use audio channels. The appendix reports additional results on the popular 1k-A (Yu et al., 2018) subset.

the increased measured performance of our method (i.e., $40.3 \rightarrow 44.7$ R@1) directly reflects the added benefits of incorporating language-based multimodal reasoning. Additionally, to keep the comparison between our method and Portillo-Quintero et al. (2021) as direct as possible, we maintain the usage of their precomputed CLIP features from ViT-B/32, but it is likely that numbers can be improved with more recent performant VLMs (e.g., LiT (Zhai et al., 2021), CLIP with ViT-L/14).

Table 5 shows that on the subset of test videos that contain *long-transcripts*, we observe a more substantial increase in performance from 40.3 to 54.9 with our method compared to Portillo-Quintero et al. (2021). Note that this is roughly comparable to the R@1 of the best *finetuned*-SOTA method, CLIP2Video (Fang et al., 2021), with 54.6 R@1 (Tab. 4). If we assume that the videos with-or-

|  | *Long-transcript subset of* MSR-VTT Full | | | |
|---|---|---|---|---|
|  | R@1↑ | R@5↑ | R@10↑ | MdR↓ |
| Portillo-Quintero et al. (2021) | 41.5 | 69.6 | 77.4 | 2.0 |
| SMs (ours) | **54.9** | **74.0** | **79.9** | **1.0** |

Tab. 5: Video-to-text retrieval on the MSR-VTT subset of videos which *long-transcripts* are available ($n=1{,}007$ of 2,990).

without transcripts are of roughly equal difficulty from a visual-only retrieval perspective, this suggests that on Internet videos with sufficient speech present in the audio, a zero-shot prompted engineered system can be competitive with *finetuned*-SOTA methods for video-to-text retrieval.

## 5 ADDITIONAL ZERO-SHOT APPLICATIONS

We describe multimodal prompt engineered systems for (i) egocentric perception, (ii) multimodal assistive dialogue, and (iii) robot perception and planning. These applications involve processing natural language inputs/feedback, live in domains for which data collection is difficult, and also serve as examples of integrating external modules (e.g., web search, robot policies) as part of the SM graph. These systems may serve as zero-shot baselines in future, when benchmarks are available to better quantitatively evaluate these capabilities (e.g., long-form egocentric video understanding, interactive multimodal assistive dialogue, and open-ended dialogue with manipulation robots). Meanwhile, in Section 6 we investigate unsupervised evaluation.

### 5.1 EGOCENTRIC PERCEPTION: USER + VLM + LLM + ALM

We can build systems to perform various perceptual tasks on egocentric video: (i) summarizing content, (ii) answering free-form reasoning questions, (iii) and forecasting. Egocentric perception has downstream applications in AR and robotics, but remains challenging: the majority of Internet-scale vision datasets focus on third-person views (Deng et al., 2009; Lin et al., 2014; Sharma et al., 2018), from which it can be difficult to transfer knowledge into the first-person domain (Li et al., 2021b; Sigurdsson et al., 2018). Further, existing egocentric datasets (Grauman et al., 2021; Damen et al., 2020; Sigurdsson et al., 2018) do not focus on long-form video comprehension and summarization. See Appendix C for an extended discussion on background in egocentric perception.

For open-ended reasoning, our SM-based system formulates *video understanding as reading comprehension*, i.e., re-framing "video Q&A" as a "short story Q&A" problem, which differs from common paradigms (see Patel et al. (2021) for a recent survey). We extract "key moments" from the video (e.g., via importance sampling, or video/audio query-based search, see Appendix), then caption the key frames using prompts similar to those in Sec. 4.1 – see examples in Fig. 4. We then recursively summarize (Wu et al., 2021b) them into a language-based record of events, a *language-based world-state history*. This is then passed as context to an LLM to perform various reasoning tasks via text completion.

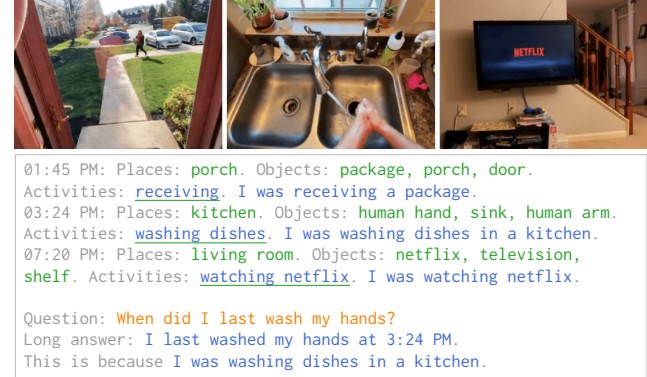

Fig. 4: SMs with VLM, LLM, and ALM can be prompted to generate captions for key moments in videos, which can be assembled into a language-based world-state history (e.g., in the form of an event log) that the LLM can answer free-form questions about.

**(i) Summarization,** given world-state history constructed using a first-person POV video[1], can be implemented by prompting an LLM to complete: "{world-state history} Summary of my day:" to which it can respond with outputs like "I slept in a bed, made coffee, watched TV, did laundry, received a package, bench pressed, showered, ate a sandwich, worked on a computer, and drank wine." **(ii) Open-ended Q&A** involves prompting the LLM to complete the template: "{world-state history} Q: {question} A:". Conditioned on the quality (comprehensiveness) of the world-state history, LLMs can generate surprisingly meaningful results to contextual recall and temporal reasoning questions questions (e.g., "what was I doing outdoors?" → "I was chopping wood in a yard.", "did I drive today?" → "no, I did not drive today.", "when did I last drink cof-

```
1:46 PM: I am eating a sandwich in a kitchen.
2:18 PM: I am checking time and working on a
laptop in a clean room. 2:49 PM: I am buying
produce from a grocery store or market.
3:21 PM: I am driving a car.
4:03 PM: I am in a park and see a playground.
4:35 PM: I am in a home and see a television.
```

Fig. 5: Example forecasting completions.

---

[1]Examples on https://youtu.be/-UXKmqBPk1w used with permission from Cody Wanner.

fee?" → "I last drank coffee at 10:17 AM", why did I go to the front porch today?" → "I went to the front porch today to receive a package."). As in (Yang et al., 2021) we can also further prompt the LLM to *explain the answer* by adding "This is because:" (Fig.4). **(iii) Forecasting** future events can be formulated as language-based world-state *completion*. Completion results (Fig. 5) are generative, and are more broad than binary event classification (Lei et al., 2020).

## 5.2 MULTIMODAL ASSISTIVE DIALOGUE: USER + VLM + LLM + WEB SEARCH

We can build an SM system to engage in multimodal dialogue to assist people in doing every day tasks, such as cooking. Our example application here helps the user search for a recipe, then guides them through it step by step. The system allows the user to navigate recipe steps with casual dialogue, provides ingredient replacements or advice (using LLM priors), and searches for visual references (in the form of images or videos) on user request. This is a case study in (i) prompting a dialogue LLM (Thoppilan et al., 2022) to produce key phrases that elicit specific multimodal interactions (e.g., video search via a VLM to output visual data), and (ii) using a web crawler (outputs in magenta) as an additional module to retrieve information online. See Appendix for the prompt. Predicted phrases "(image)" initiates the web crawler to return a recipe video (if it exists), and the VLM runs visual search on it using the LLM response to retrieve related frames e.g., [2]:

```
Alice: Step: In a medium bowl, beat the egg whites until frothy.
Bob: what do you mean by medium bowl.
Alice: A medium bowl is a bowl that is big enough to fit the ingredients for this recipe and
that is comfortable to mix in.
Bob: exactly how big should that be.
Alice: (image) A medium bowl is a bowl that is big enough to fit the ingredients for this
recipe and that is comfortable to mix in.
(a) video search {recipe video} with "A medium bowl is a bowl that is..."
Bob: ok assume im done whats next.
Alice: Step: Keep beating and slowly add the sugar until stiff peaks form.
Bob: whats stiff peaks.
Alice: Stiff peaks are when egg whites stand up in peaks that do not fall over.
Bob: can you show me what that looks like.
Alice: (image) Stiff peaks are when egg whites stand up in peaks that do not fall over.
(b) video search {recipe video} with "Stiff peaks are when the egg whites..."
```

Fig. 6: Multimodal prompt engineering VLM, Web Search, and LLM can enable multimodal dialogue applications such as guiding a user through online recipe steps and providing assistive visuals via video search.

## 5.3 ROBOT PERCEPTION & PLANNING: USER + VLM + LLM + POLICIES

SM-based systems can be used to enable robots to perform language-conditioned tasks. Our example uses a VLM (open vocabulary object detection with ViLD (Gu et al., 2021)) to describe objects in the scene, feeds that description as context to a LLM as a multi-step planner (Ahn et al., 2022; Huang et al., 2022), that then generates the individual steps to be passed to a pretrained language-conditioned robot policy (e.g., models similar to CLIPort (Shridhar et al., 2022; Zeng et al., 2020) for open vocabulary pick-and-place). Steps can be represented in the form of natural language ("Pick the red block and place it on the blue block.") or in the form of pseudocode (to generate text with a fixed template e.g., "robot.pick_and_place("red block", "blue block")"), leveraging LLM capacity to write code. We show this in the context of a simulated environment (shown in Fig. 7) using a UR5 arm and and several objects (blocks, bowls). Distinct from Ahn et al. (2022), this uses VLM-informed in-context substitution and LLM code generation, rather than joint probabilistic inference.

```
objects = ["green block", "blue block", "yellow block", "green
bowl", "blue bowl", "yellow bowl"]
# stack the blocks on top of each other.
Step 1. robot.pick_and_place("yellow block", "blue block")
Step 2. robot.pick_and_place("green block", "yellow block")
# wait actually undo that last step.
Step 1. robot.pick_and_place("green block", "top left corner")
# put the yellow block in the bowl you think it best fits.
Step 1. robot.pick_and_place("yellow block", "yellow bowl")
```

Fig. 7: Multimodal prompt engineering VLM, LLM, and language-conditioned policies (via CLIPort (Shridhar et al., 2022)) can enable robots to parse and generate plans from free-form human instructions (in orange).

---

[2]Example using recipe steps and ingredients from tasty.co/recipe/strawberry-cheesecake-macarons

Chaining this system together expands the set of language-specified tasks beyond the original set of primitives trained by the policy, and enables applications involving human dialogue with the robot.

# 6 UNSUPERVISED EVALUATION FOR MODEL SELECTION

The zero-shot application of SM systems without training data (as in Sec. 5) raises an interesting question which we investigate – how do we evaluate models? To address this, we extend (Strope et al., 2011) (originally used for speech recognition) with a visual-language case study for the application of generating world-state histories (Sec. 5.1), although the method could be adapted for other multimodal tasks as well. Since our metric of interest is the *combined* performance of e.g., a VLM and a LLM – rather than asking the question: '(A): how well does a VLM perform in absolute?' for SMs, we can instead ask: '(B): how well does this VLM compensate for the weaknesses of the LLM?'. (Strope et al., 2011) show that answering (B) correlates well with answers (A), and is useful e.g., for model selection. Specifically, to evaluate a new VLM' for generating language-based world-state history, we first use a baseline VLM paired with the strong LLM (sLLM) to generate pseudo ground truth predictions VLM×sLLM. We then take both the baseline VLM and new VLM', and pair them with a weak LLM wLLM to generate predictions VLM× wLLM and VLM'×wLLM respectively. We score these predictions by similarity to the pseudo ground truth VLM×sLLM. This can be done by by using a similarity-scoring language model e.g., RoBERTa (Liu et al., 2019b).

Tab. 6 shows example results of this analysis with GPT-3 "davinci" as the sLLM, and "curie" as the wLLM, to compare VLM (i.e., CLIP) variants with different backbones: vision transformers (ViT) (Dosovitskiy et al., 2020) and ResNets (RN50) (He et al., 2016) with different model sizes. We find that this method can capture a moderate correlation of ascending performance with increasingly better VLMs (e.g., better variants of CLIP) (Radford et al., 2021), as measured by zero-shot image classification accuracy on ImageNet (Deng et al., 2009) – with correlation coefficients of 0.41 and 0.46 between ImageNet accuracies and mean similarity to

| | VLM (CLIP) Variants + Weak LLM | | | | |
|---|---|---|---|---|---|
| Truth Models | RN50x4 | RN50x16 | ViT-B/32 | ViT-B/16 | ViT-L/14 |
| GPT-3 + ViT-B/16 | 0.628 | 0.646 | 0.686 | 0.861 | **0.704** |
| GPT-3 + RN50x16 | 0.667 | 0.851 | 0.689 | 0.655 | **0.704** |
| ImageNet Accuracy | 65.8 | 70.5 | 63.2 | 68.6 | 76.2 |
| Size (# params) | 178M | 291M | 151M | 150M | 427M |

Tab. 6: Unsupervised evaluation (higher is better) of various VLMs by pairing them with a weak LLM and comparing outputs to a VLM paired with a strong LLM, which provides relative 'truth gradients' that inform how well the VLMs can compensate for the weak LLM. These results show that the best VLM for this SM system correlates with the best zero-shot ImageNet model.

truth models via ViT-B/16 and RN50x16 respectively. Specifically for our SM egocentric perception systems, model combinations that use the same VLM as the one that generates ground truth are biased to produce similar visual grounding results and can exhibit an unfair advantage during the comparisons. Those numbers have been grayed out in Tab. 6.

# 7 LIMITATIONS AND DISCUSSION

SM systems leverage in-context multimodal prompt engineering as a means to compose multiple large pretrained models to make predictions for new multimodal tasks, that each model may otherwise struggle to do independently. These systems can (i) serve as strong zero-shot baselines that are competitive with state-of-the-art on standard multimodal benchmarks, (ii) adapt large pretrained models for multimodal tasks while retaining their robustness to distribution shifts (known to deteriorate after finetuning (Wortsman et al., 2021)), and (iii) present practical application advantages in domains that are restricted by data scarcity, training compute, or model access. Such systems are however, subject to inheriting the limitations of the models on which they are built. For example, our captioning systems use VLMs (e.g., CLIP) predominantly as zero-shot image classifiers, so the extent to which they can express visual information is more contextual than fine-grained. However, we expect that by replacing the VLMs with more expressive ones (e.g., ViLD (Gu et al., 2021), LSeg (Li et al., 2022a), or Flamingo (Alayrac et al., 2022)), SMs may likewise benefit in the capacity to express details. Future work may also involve meta-learning the multimodal exchanges themselves, expanding the intermediate representations from discrete to continuous (Gal et al., 2022), or extending them to include additional modalities beyond text, e.g., passing images between modules.

**Reproducibility statement.** SM-based systems are in part by nature, simple and easy to reproduce. We provide open-source implementations with code that can be directly run in the browser with Colab. See https://socraticmodels.github.io for download links.

**Ethic statement.** Multimodal systems with natural language as middleware provides an interpretable window into the behavior of the systems (even for non-experts). The barrier of entry is small: SMs can be engineered to capture new functionalities with minimal additional resources, and tackle applications that have traditionally been data-scarce. This can be enabling, but also raises potential risks, since it increases the flexibility of unintended end use applications, and should be carefully monitored over time. It is also important to note that the system may generate results that reflect unwanted biases found in the Internet-scale data on which incorporated models are trained, and should be used with caution (and checked for correctness) in downstream applications. We welcome broad discussion on how to maximize potential positive impacts (enabling broad, new multimodal applications, with minimal resources) while minimizing the capabilities of bad actors.

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

# Appendix for *Socratic Models*

## A  OVERVIEW

The appendix includes: (i) additional notes on main experiments, (ii) more details on applications to egocentric perception, (iii) scaling video search for language-based world-state history, (iv) full outputs for multimodal assistive dialogue, (v) more details on robot perception and planning experiments, (vi) additional discussion on future work (e.g., SMs for deductive reasoning) and (vii) broader impacts (e.g., energy and resource consumption). For code, see https://socraticmodels.github.io.

## B  ADDITIONAL NOTES ON EXPERIMENTS

**Model choices.** There are many options of large pretrained "foundation" (Bommasani et al., 2021) models to choose from, but our experiments in the main paper use models that are publicly available, so that our systems can be made accessible to the community. In particular, we use CLIP (Radford et al., 2021) as the text-image similarity VLM (ViT-L/14 with 428M params, except on MSR-VTT which uses ViT-B/32), ViLD (Gu et al., 2021) as the open-vocabulary object detector VLM; Wav2CLIP (Wu et al., 2021a) as the sound-critic ALM and Google Cloud Speech-to-text API (gcl) as the speech-to-text ALM; GPT-3 with 175B params (Brown et al., 2020; Ouyang et al., 2022) and RoBERTa (Liu et al., 2019b) with 355M params as the LLMs. All pretrained models are used off-the-shelf with no additional finetuning. In terms of compute resources required, all experiments can be run on a single machine using an NVIDIA V100 GPU with internet access for outsourced API calls (e.g., GPT-3 and Google Cloud Speech-to-text).

### B.1  IMAGE CAPTIONING ON MS COCO

For image captioning experiments on the MS COCO dataset (Chen et al., 2015; Lin et al., 2014), we evaluate over a random sampled subset of 100 images from the test split (Karpathy & Fei-Fei, 2015), so that GPT-3 API runtime costs are more affordable for reproducibility ($\sim$\$150 USD per run with with $n = 20$ generated candidate captions per image).

| Method | BLEU-4 | METEOR | CIDEr | SPICE | ROUGE-L |
|---|---|---|---|---|---|
| [†] MAGIC (Su et al., 2022) (full) | 12.9 | 17.4 | 49.3 | 11.3 | 39.9 |
| ZeroCap (Tewel et al., 2021) (full) | 2.6 | 11.5 | 14.6 | 5.5 | – |
| [*] ClipCap (Mokady et al., 2021) (full) | 33.5 | 27.5 | 113.1 | 21.1 | – |
| [†] MAGIC (Su et al., 2022) (subset) | 11.4 | 16.4 | 56.2 | 11.3 | 39.0 |
| ZeroCap (Tewel et al., 2021) (subset) | 0.0 | 8.8 | 18.0 | 5.6 | 18.3 |
| [*] ClipCap (Mokady et al., 2021) (subset) | 40.7 | 30.4 | 152.4 | 25.2 | 60.9 |

[*] finetuned on full training set with image-text pairs.
[†] finetuned on unpaired training set, zero-shot on image-text pairs.

Tab. 7: Image captioning metrics on the random subset of $N = 100$ (bottom) test examples are comparable to the full MS COCO test set metrics (top).

While test performance on a smaller subset may exhibit more variation, the metrics shown in Tab. 7 from the baselines reported on this subset of MS COCO test examples are a close approximation of their full test set metrics. Also, while the captions in Fig. 3, Section 4.1, were generated with the prompt "...creative short..." as noted in Fig. 3, for best quantitative MS COCO captions we used the prompt "...short, likely...". The subset of MS COCO image IDs are as follows:

```
002239 003001 004497 015660 018534 019176 020254 020779 022324 025145
025394 028714 028850 034567 054562 055299 055694 083049 086877 097465
098872 108169 111811 113040 126260 127161 131969 147725 166482 173932
177941 183144 187450 187979 195645 199442 205232 216867 223540 226802
235057 237118 240655 242297 246562 246955 255274 257965 259342 271266
273909 279149 296865 301494 315908 317244 320893 330265 340179 347648
357493 358268 365444 368242 370208 374990 384670 400538 406189 417455
419408 430690 435682 449976 462527 463730 465007 465550 468018 471350
472376 472904 483135 486805 494887 500657 509750 513497 515993 526392
533171 542922 552612 553698 558594 559656 564095 565683 566502 568508
```

**Few-shot prompting.** For image captioning, we concatenate a few ground-truth caption training examples with VLM predictions and add them to the prompt, as is done similarly for standard few-shot language model examples. We find that captioning performance tends to asymptote with more than 3 random training examples (we tested 8 as well, results

| Method | BLEU-4 | METEOR | CIDEr | SPICE | ROUGE-L |
|---|---|---|---|---|---|
| SMs 0-shot (ours) | 10.0 | 16.2 | 50.1 | 10.8 | 36.1 |
| SMs 3-shot (ours) | **18.2** | **20.5** | **76.3** | 13.9 | **43.7** |
| SMs 8-shot (ours) | 15.4 | 20.4 | 73.0 | **14.9** | 43.1 |

Tab. 8: SMs improve on MS COCO image captioning with few-shot prompted examples, but tends to asymptote on performance with more than 3 examples.

shown in Tab. 8), though future work may investigate dynamic prompts to retrieve few-shot examples that maximize test performance. Zero-shot prompt:

```
I am an intelligent image captioning bot. This image is a {img_type}. There {num_people}. I think this photo
was taken at a {place1}, {place2}, or {place3}. I think there might be a {object1}, {object2}, {object3},...
in this {img_type}. A short, likely caption I can generate to describe this image is:
```

One-shot prompt:

```
I am an intelligent image captioning bot. This image is a {img_type}. There {num_people}. I think this photo
was taken at a {place1}, {place2}, or {place3}. I think there might be a {object1}, {object2}, {object3},...
in this {img_type}. A short, likely caption I can generate to describe this image is: Three friends sitting on
the beach together.

I am an intelligent image captioning bot. This image is a {img_type}. There {num_people}. I think
this photo was taken at a {place1}, {place2}, or {place3}. I think there might be a {object1}, {object2},
{object3},... in this {img_type}. A short, likely caption I can generate to describe this image is:
```

Tab. 9 shows additional ablations compared to greedy search, and nucleus sampling (which performs favorably over beam search for captioning tasks according to Li et al. (2022b)). We observe that sampling with temperature performs comparably (slightly better for most metrics) versus both alternatives.

| Method | BLEU-4 | METEOR | CIDEr | SPICE | ROUGE-L |
|---|---|---|---|---|---|
| N=20 Temp=0.9 Sampling (original) | **10.0** | 16.2 | **50.1** | **10.8** | 36.1 |
| N=20 Nucleus Sampling (top_p=0.9) | 8.5 | **16.6** | 47.7 | 10.8 | 36.4 |
| N=1 Greedy Search | 9.3 | 15.9 | 49.5 | 10.5 | **36.9** |

Tab. 9: Generating captions with SMs performs comparably with different LLM decoding schemes as alternatives.

### B.2 CONTEXTUAL IMAGE CAPTIONING ON CONCADIA

Our experiments on Concadia (Kreiss et al., 2021) evaluate the extent to which SMs can generate captions and descriptions conditioned on input images and their associated article text. While our results show that the SM combination of VLMs and LLMs can achieve strong results on the benchmark, we also observe that LLMs (e.g., GPT-3) alone can return surprisingly competitive results too (Tab. 10). Specifically, using the same LLM prompt but leaving out information from the VLM:

```
I am an intelligent image captioning bot. The article is about: "{article_text}". In this image, I think I see
a {object1}, {object2}, {object3},... A short caption for this image is:
```

subsequently drops CIDEr performance on image description, but slightly improves captioning. This suggests: (i) information from VLM is more important for LLMs in generating descriptions than captions, (ii) there may be strong correlation between distributions of captions and article texts that can be leveraged by an LLM alone, and/or (iii) there may exist overlap between Concadia (e.g., Wikipedia text) and the training set of the LLM, which warrants further investigation to disentangle confounding variables.

| Method | Caption Generation | Description Generation |
|---|---|---|
| Kreiss et al. (Kreiss et al., 2021) | 11.3 | 17.4 |
| SMs (ours) | 38.8 | 23.0 |
| **SMs (no image)** | 40.1 | 20.6 |
| SMs w/ description | 93.8 | – |

Tab. 10: SMs on zero-shot contextual image captioning and description tasks on the Concadia dataset.

For prompt engineering, replacing the prompt from "intelligent image captioning bot" to "intelligent image describing bot" appears to slightly reduce performance (results in Tab. 11). Interestingly, we observe that the generated descriptions are quite similar to those generated with "captioning" in the prompt – and while more

| Prompt Changes | Description Generation |
|---|---|
| "image captioning bot" (original) | 23.0 |
| "image describing bot" | 21.3 |

Tab. 11: Zero-shot SMs are subject to subtle changes in the wording of LLM prompts.

verbose in some cases, do not necessarily add new information on visual details. It may be interesting future work to explore additional ways to extract perceptual cues from the VLM.

### B.3 VIDEO-TO-TEXT RETRIEVAL ON MSR-VTT 1K-A

We also report results in Tab. 12 on the popular MSR-VTT "1k-A" subset, introduced by (Yu et al., 2018) created via random sampling on the full test set. We follow the same evaluation protocol for video-to-text retrieval as used in prior work (Liu et al., 2019a; Dong et al., 2019; 2018; Mithun et al., 2018), which reports the minimum rank among all valid text captions for a given video query, and each test video is associated with 20 captions.

| Category | Method | MSR-VTT 1k-A | | | | Audio | CLIP enc. |
|---|---|---|---|---|---|---|---|
| | | R@1↑ | R@5↑ | R@10↑ | MdR↓ | | |
| | Collaborative Experts (Liu et al., 2019a) | 20.6 | 50.3 | 64.0 | 5.3 | yes | |
| | SSB (Patrick et al., 2020) | 28.5 | 58.6 | 71.6 | 3.0 | no | |
| | CLIP4Clip (Luo et al., 2021) | 43.1 | 70.5 | 81.2 | 2.0 | no | ViT-V/32 |
| | CLIP2Video (Fang et al., 2021) | 43.5 | 72.3 | 82.1 | 2.0 | no | ViT-V/32 |
| *Finetuning* | DRL (Wang et al., 2022), ViT-B/32 | 45.3 | 73.9 | 83.3 | 2.0 | no | ViT-V/32 |
| | CAMoE (Cheng et al., 2021) | 49.1 | 74.3 | 84.3 | 2.0 | no | ViT-B/32 |
| | CLIP2TV (Gao et al., 2021) | 54.1 | 77.4 | 85.7 | **1.0** | no | ViT-B/16 |
| | DRL (Wang et al., 2022), ViT-B/16 + QB-n | **56.2** | **79.9** | **87.4** | **1.0** | no | ViT-B/16 |
| | SSB (Patrick et al., 2020), zero-shot | 8.7 | 23.0 | 31.1 | 31.0 | no | |
| *Zero-shot* | CLIP via (Portillo-Quintero et al., 2021) | 58.0 | 82.5 | 90.2 | 1.0 | no | ViT-B/32 |
| | SMs (ours) | **60.7** | **84.1** | **90.6** | **1.0** | yes | ViT-B/32 |

Tab. 12: Video-to-text retrieval results on MSR-VTT (Xu et al., 2016) dataset on the 1k-A (Yu et al., 2018) subset. Differentiated are methods which train on the MSR-VTT dataset (*finetuning*), compared with *zero-shot* methods, which do not. Also noted: whether the methods use audio channels, and if CLIP (Radford et al., 2021) is used, which CLIP encoder is used.

Note that the original CLIP baseline for video-to-text retrieval via Portillo-Quintero et al. (Portillo-Quintero et al., 2021) reports R@1 to be 27.2, but this was computed with only 1 caption per video that was random sampled (Yu et al., 2018) from the original set of 20 captions (for text-to-video retrieval). This differs from the original evaluation protocol and may be sub-optimal since the sampled caption can be ambiguous or partial (generated from crowd compute). For example, videos may be paired with a vague caption "a person is explaining something" as ground truth, rather than one of the other (more precise) captions e.g., "a person is talking about importing music to a ipod". Upon correcting the evaluation protocol (i.e., increasing the number of associated captions per video to 20), R@1 for Portillo-Quintero et al. (Portillo-Quintero et al., 2021) improves to 58.0, and SMs improve on top of that with LLMs and ALMs[3] to 60.7 R@1 zero-shot.

Other methods have also evaluated on zero-shot MSR-VTT *text-to-video* retrieval (Xu et al., 2021; Miech et al., 2020; Bain et al., 2021), but these have all been outperformed by Portillo-Quintero et al. (Portillo-Quintero et al., 2021). Our method may be adapted as well to *text-to-video*, but due to our use of transcripts on only a subset of the videos, unlike in video-to-text, this creates an asymmetry which may require an unwieldy relative weighting for ranking videos with or without transcripts. Note that (Tab. 12) prior to the CLIP revolution in video-to-text retrieval, using the audio modality was not uncommon amongst competitive video-to-text retrieval methods (Mithun et al., 2018; Liu et al., 2019a). The trend over the past year, however, has been to instead focus on using only visual features, with *all* recent competitive methods being based off of CLIP, and not using audio data. Our approach, through leveraging commonsense reasoning stored in the LLMs, is able to once again allow audio data to enable progress in this common video understanding task, beyond what CLIP alone can provide.

Fig. 8 shows a diagram of the SM systems used for image captioning and video-to-text retrieval.

## C EGOCENTRIC PERCEPTION APPENDIX

**Background.** Egocentric perception continues to be an important problem in computer vision. Early work in the area explores hand-designed first-person visual features for egocentric action

---

[3] Key used parameters for Google Cloud Speech-to-Text API include 'model=video' and 'use_enhanced=True'. At 0.006 cents per 15 seconds, this represents an estimated speech-to-text processing cost of under 25 cents (USD) for all MSR-VTT test data.

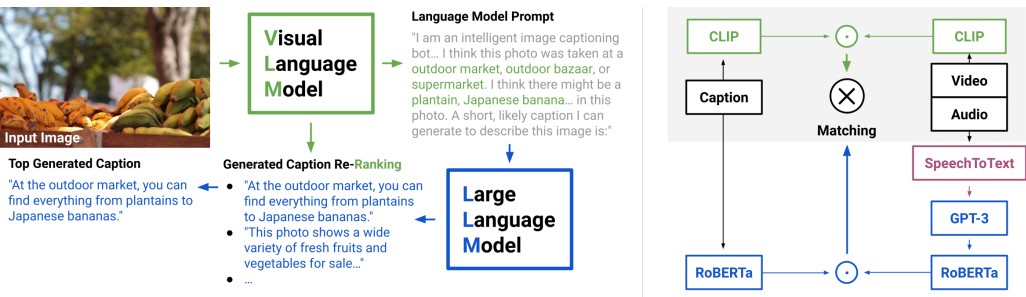

Fig. 8: Flow diagrams of how VLMs, LLMs, ALMs can be combined together as systems for image captioning tasks (left) and video-to-text retrieval (right). Note that SMs can be used to augment Portillo-Quintero et al. (2021) for video-to-text retrieval (highlighted in gray) with additional information from audio modalities.

recognition, object understanding, and video summarization. This includes ego-motion (e.g., optical flows) (Kitani et al., 2011; Ryoo & Matthies, 2013) as well as features from human gaze, hands, and objects (Spriggs et al., 2009; Lee et al., 2012; Fathi et al., 2011; Pirsiavash & Ramanan, 2012; Li & Kitani, 2013; Lee & Grauman, 2015). Focusing on hand-designed features was common in early egocentric vision research, as the availability of data (or videos in general) was very limited. More recent approaches in egocentric perception leverage learned feature representations, utilizing pretrained convolutional network features (Ryoo et al., 2015), finetuning them (Ma et al., 2016; Zellers et al., 2022), or training them from scratch (Bambach et al., 2015) with first-person videos. Similar to the topics explored in early work, learning of visual representations capturing human hands, objects, and eye gaze has been extensively studied (Garcia-Hernando et al., 2018; Li et al., 2018). (Kazakos et al., 2019) learns multimodal embeddings (i.e., video + audio), and (Furnari & Farinella, 2019) studies future action anticipation from egocentric videos. Lack of sufficient data however, consistently remains a bottleneck – motivating researchers to construct new larger-scale egocentric video datasets including EPIC-Kitchens (Damen et al., 2018), Charades-Ego (Sigurdsson et al., 2018), and Ego4D (Grauman et al., 2021).

## C.1 WHY EGOCENTRIC PERCEPTION?

We highlight SMs on egocentric perception because it is an important yet challenging computer vision domain (Grauman et al., 2021; Damen et al., 2020; Sigurdsson et al., 2018) with downstream applications in augmented reality (AR) and robotics (Ahn et al., 2022). From unusual viewpoints to the lack of temporal curation – the characteristics of first-person videos are unique and not often found in existing datasets, which focus more on generic Internet content captured from third-person spectator views (Deng et al., 2009; Lin et al., 2014; Sharma et al., 2018). Notably, this domain shift makes it difficult for data-driven egocentric models to benefit from the standard paradigm of pretraining on third person Internet data (Li et al., 2021b; Sigurdsson et al., 2018). Overall, the key challenges have included how to acquire sufficient egocentric data, and/or how to make sufficient use of this data (either with dense labels, or otherwise).

Despite the challenges of egocentric perception, we find that SMs can reconcile the complementary strengths of pretrained foundation models to address these difficulties through contextual reasoning. For example, while modern activity recognition models trained on third person data might over-index to the motion of the primary person in video (making the models difficult to be adapted to first-person videos), we find that LLMs like GPT-3 can suggest equally plausible activities (e.g., "receiving a package") that may be occurring given only a brief description of the scene (e.g., "front porch") and the objects detected in the image ("package, driveway, door") by a VLM. These activity suggestions are often more expressive than the class categories that can be found in typical activity recognition datasets (e.g., Charades (Sigurdsson et al., 2018), Kinetics (Smaira et al., 2020)), and reflect the information already stored in the models, agnostic to the point of view. Our SM system for egocentric perception leverages these advantages, and also suggests future research directions in contextual reasoning that leverage existing language-based models without having to curate large annotated datasets.

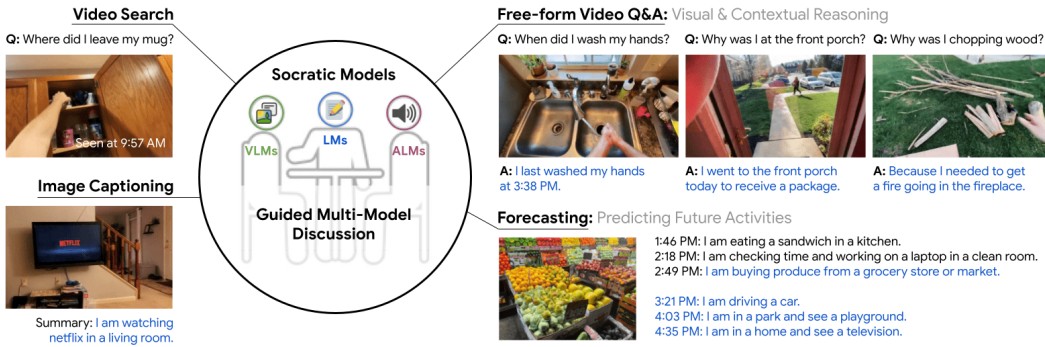

Fig. 9: On various egocentric perceptual tasks (shown), this work presents a case study of SMs with visual language models (VLMs, e.g., CLIP), large language models (LMs, e.g., GPT-3, RoBERTa), and audio language models (ALMs, e.g., Wav2CLIP, Speech2Text). From video search, to image captioning; from generating free-form answers to contextual reasoning questions, to forecasting future activities – SMs can provide meaningful results for complex tasks across classically challenging computer vision domains, without any model finetuning.

## C.2 Additional Details on Language-Based World-State History from Video

In order to provide language-based reasoning capabilities for open-ended question-answering, a key aspect of our system is to describe the observed states of the world in language, with the goal of creating a language-based world-state history (Fig. 10) that can be used as context to an LM. To this end, a component of our method generates Socratic image summaries of individual video frames (Sec. 3.3-A), that can then be concatenated (along with timestamps) to form an event log (illustrated at the top and middle of Fig. 10).

***3.3-A. Socratic Egocentric Image Summaries.*** Given an image frame as input, this component generates a natural language summary (e.g., caption) of what is occurring in the image. Our system uses a Socratic approach with guided multimodal multi-model discussion to provide answers to 3 questions that describe the visual scene: "where am I?", "what do I see?", and "what am I doing?", which are then summarized into a single caption per image frame.

- **Where am I?** For place recognition, we use a VLM to rank Places365 (Zhou et al., 2016) scene categories against the image, with the top $n$ candidates (out of 365) inserted into a prefix: "Places: {place1}, {place2}, {place3}.".

- **What do I see?** For object and people recognition, we use a VLM to rank OpenImages object categories (Kuznetsova et al., 2020) against the image, with the top $m$ categories (out of 600) inserted into a second prefix: "Objects: {object1}, {object2}, {object3}."

- **What am I doing?** For activity recognition, we use a back-and-forth interaction between an LLM and VLM: we first use an LLM to infer the activities most related to the places and objects previously listed by the VLM (green):

```
Places: {place1}, {place2}, {place3}. Objects: {object1}, {object2}, {object3}.
Activities: activity_a, activity_b, activity_c.
```

We find that generating candidate activities using an LLM yields more suitable descriptions of egocentric activities and interactions with first-person video, than using standard activity recognition dataset categories (e.g., from Charades or Kinetics). Activity recognition datasets are often tailored to third person videos, and can only cover a partial subset of human activities, which instead can be more holistically captured through LLM reasoning (Petroni et al., 2019) over the objects and places that the VLM perceives. For example, "receiving a package" is a common household activity not found in most datasets. After the LLM generates candidate activities, these candidates are then fed back to the VLM and re-ranked to sort out the top $k$ activities by relevance to the key image frame: "Activities: {activity1}, {activity2}, {activity3}."

This process of generating candidate activities from places and objects is one way of extracting commonsense from LLMs as knowledge bases (Petroni et al., 2019). Continuing the Socratic dialogue further, this can be repeated likewise to generate new relevant objects (conditioned on activities

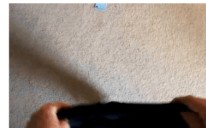 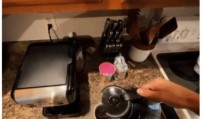 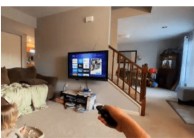 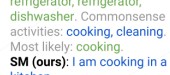 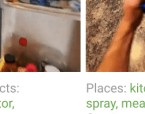

Places: clean room. Objects: shorts, jeans, shirt. Commonsense activities: getting dressed. Most likely: getting dressed. **SM (ours)**: I am getting dressed.

**ClipCap**: how to make a pair of jeans.

Places: kitchen. Objects: coffeemaker, waffle iron, kettle. Commonsense activities: making coffee, making waffles. Most likely: making coffee. **SM (ours)**: I am making coffee, waffles, and tea.

**ClipCap**: how to clean a stove with a brush.

Places: living room. Objects: remote control, television, netflix. Commonsense activities: watching netflix. Most likely: watching netflix. **SM (ours)**: I am watching netflix on the television.

**ClipCap**: this is what the house looks like from the inside.

Places: kitchen. Objects: refrigerator, refrigerator, dishwasher. Commonsense activities: cooking, cleaning. Most likely: cooking. **SM (ours)**: I am cooking in a kitchen.

**ClipCap**: the refrigerator is full of food.

Places: kitchen. Objects: cooking spray, measuring cup, mixing bowl. Commonsense activities: measuring, mixing. Most likely: mixing. **SM (ours)**: I am mixing a recipe.

**ClipCap**: how to make a mason jar with a lid.

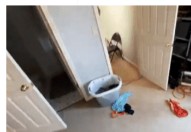 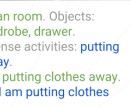 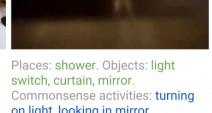 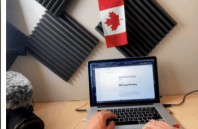 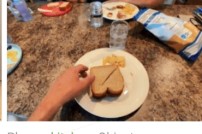 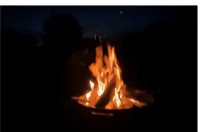

Places: clean room. Objects: closet, wardrobe, drawer. Commonsense activities: putting clothes away. Most likely: putting clothes away. **SM (ours)**: I am putting clothes away.

**ClipCap**: the dog's owner was left shocked when the cat jumped out of the way of the door.

Places: shower. Objects: light switch, curtain, mirror. Commonsense activities: turning on light, looking in mirror, showering. Most likely: showering. **SM (ours)**: I am showering and see the typical objects in a shower.

**ClipCap**: the video shows the man running away from the camera.

Places: home office. Objects: flag, poster, computer monitor. Commonsense activities: work on computer, look at flag, look at poster. Most likely: work on computer. **SM (ours)**: I am work on computer in home office.

**ClipCap**: the computer is now working on the screen.

Places: kitchen. Objects: sandwich, hamburger, kitchen & dining room table. Commonsense activities: eating, sitting. Most likely: eating. **SM (ours)**: I am eating a sandwich in a kitchen.

**ClipCap**: person, who is a student, said she was shocked when she saw the sandwich on the table.

Places: campsite. Objects: fireplace, torch, wood-burning stove. Commonsense activities: cooking, camping. Most likely: camping. **SM (ours)**: I am camping and can see a fireplace, torch, and wood-burning stove.

**ClipCap**: campfire in the night, slow motion.

📝 **Generated Language-Based World-State History from Egocentric Video**

08:31 AM: Places: clean room. Objects: shorts, jeans, shirt. Activities: getting dressed. I was getting dressed.
10:17 AM: Places: kitchen. Objects: coffeemaker, waffle iron, kettle. Activities: making coffee. I was making coffee , waffles, and tea.
11:09 AM: Places: living room. Objects: remote control, television, netflix. Activities: watching netflix. I was watching netflix on the television.
01:17 PM: Places: staircase. Objects: stairs, hamster, human leg. Activities: ascending. I was ascending a staircase and see a hamster on the stairs and a human leg.
01:45 PM: Places: porch. Objects: package, porch, door. Activities: receiving. I was receiving a package.
03:24 PM: Places: kitchen. Objects: human hand, sink, human arm. Activities: washing dishes. I was washing dishes in a kitchen.
03:38 PM: Places: kitchen. Objects: refrigerator, refrigerator, dishwasher. Activities: cooking. I was cooking in a kitchen.
03:52 PM: Places: kitchen. Objects: cooking spray, measuring cup, mixing bowl. Activities: mixing. I was mixing a recipe.
04:24 PM: Places: shower. Objects: light switch, curtain, mirror. Activities: showering. I was showering and see the typical objects in a shower.
04:35 PM: Places: home office. Objects: flag, poster, computer monitor. Activities: work on computer. I was work on computer in home office.
05:16 PM: Places: kitchen. Objects: sandwich, hamburger, kitchen & dining room table. Activities: eating. I was eating a sandwich in a kitchen.
05:27 PM: Places: clean room. Objects: laptop, digital clock, desk. Activities: checking time. I was checking time and working on a laptop in a clean room.
06:31 PM: Places: field road. Objects: girl, worm, centipede. Activities: walking. I was walking in a field and see a girl and a worm.
07:01 PM: Places: yard. Objects: axe, axe, chainsaw. Activities: chopping wood. I was chopping wood.
07:12 PM: Places: campsite. Objects: fireplace, torch, wood-burning stove. Activities: camping. I was camping and can see a fireplace, torch, and wood-burning stove.

🔁 **Open-Ended Reasoning**

**Recall Q&A**
**Q**: Did I eat dinner today? **A**: Yes, I ate dinner today. **Explanation**: I was seen eating a sandwich in a kitchen at 5:27 PM.
**Q**: Did I drive today? **A**: No, I did not drive today. **Explanation**: I was at home all day.
**Q**: Did I do laundry today? **A**: No, I did not do laundry today. **Explanation**: I was putting clothes away in a closet and wardrobe and drawer in a clean room.

**Contextual Reasoning Q&A**
**Q**: What was I doing outdoors? **A**: I was chopping wood in a yard. **Explanation**: I need wood to keep my fireplace going.
**Q**: Why was I chopping wood? **A**: I was chopping wood because I needed to get a fire going in the fireplace. **Explanation**: I was camping and it was getting cold.
**Q**: Why did I go to the front porch today? **A**: I went to the front porch today to receive a package. **Explanation**: I saw on the porch a package and knew that I was expecting it.
**Q**: Why were my wife and I in the living room this evening? **A**: My wife and I were in the living room this evening because we were watching netflix. **Explanation**: we have a television in the living room and netflix is a service that we can watch on it.

**Temporal Reasoning Q&A**
**Q**: When did I last wash my hands? **A**: I last washed my hands at 3:38 PM. **Explanation**: I was washing dishes in a kitchen.
**Q**: How many times did I receive a package today? **A**: I received a package once today. **Explanation**: I was receiving a package at 03:24 PM.
**Q**: When did I last drink coffee? **A**: I last drank coffee at 10:17 AM. **Explanation**: I was making coffee in the kitchen.
**Q**: When did I last watch Netflix? **A**: I last watched Netflix at 7:32 PM on April 6, 2020. **Explanation**: I was watching Netflix in a living room.
**Q**: How many times did I use electronics today? **A**: I used electronics six times today. **Explanation**: I was working on a laptop in a clean room, checking time and watching netflix on a television, and streaming netflix on a computer monitor.
**Q**: When did I take a shower? **A**: I took a shower at 4:35 PM. **Explanation**: I was showering and see the typical objects in a shower.
**Q**: How long did I watch TV today? **A**: I watched TV for about 5 hours today. **Explanation**: I was watching netflix on the television for about 3 hours and then I watched a woman drink w in a living room for about 2 hours.

**Subjective Q&A**
**Q**: Was I happy today? **A**: I was generally happy today. **Explanation**: I was able to do some activities that I enjoy, such as watching netflix, working on my computer, and eating a sandwich.
**Q**: What was my favorite drink today? **A**: I drank wine in a living room with a woman. **Explanation**: I like to drink wine with friends.

Fig. 10: An instantiation of the SMs framework for open-ended reasoning with egocentric perception. SMs can generate meaningful structured captions (top) for egocentric images through Socratic dialogue between VLMs (green) and LLMs (blue), and qualitatively perform well versus state-of-the-art captioning models such as ClipCap (Mokady et al., 2021). Key moments from egocentric video are summarized with SMs into a language-based world-state history (middle), which can be provided as context to an LLM for open-ended question answering. Results (bottom) for generated answers (blue) and model explanations (blue) suggest SMs are fairly capable of performing a variety of reasoning tasks including answering binary yes or no questions, contextual and temporal reasoning questions, as well as subjective questions.

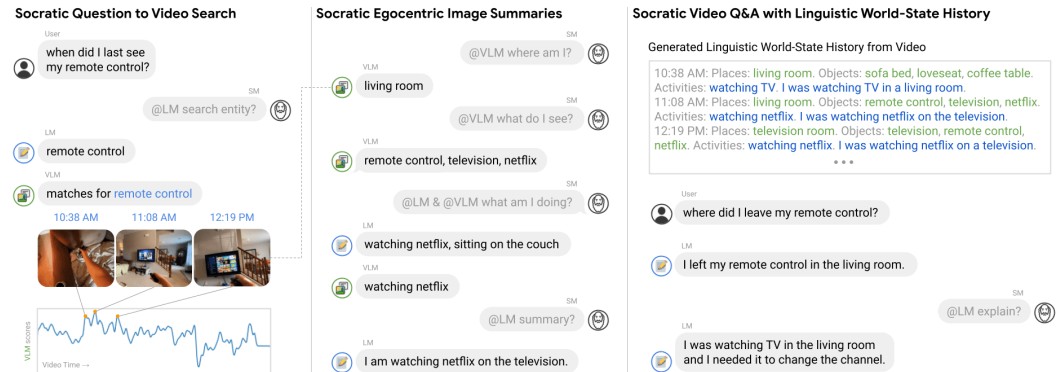

Fig. 11: Examples of guided multi-model exchanges (Socratic Models) for an egocentric perception system: (i, left) parsing a natural language question into search entities (with LLM) to be used to find the most relevant key moments in the video (with VLM); (ii, middle) describing each key frame by detecting places and objects (VLM), suggesting commonsense activities (LLM), pruning the most likely activity (VLM), then generating a natural language summary (LLM) of the SM interaction; (iii, right) concatenating key frame summaries into a language-based world-state history that an LLM can use as context to answer the original question.

and places), as well as new places (conditioned on objects and activities). One can iterate the procedure (LLM generate, VLM re-rank, repeat) to populate the set of places, objects, and activities until equilibrium (i.e., no more new entities), which generally helps to cover a broader set of places and objects that expand beyond the initial seed categories from Places365 and OpenImages. For example:

```
If I am making making pancakes, objects that I am likely to see include: a frying pan,
a spatula, a bowl, milk, eggs, flour, sugar, baking powder, butter, a plate, syrup.
```

Given the final set of places, objects, and activities, we use the LLM to generate an overall first-person summary of what is happening in the image. Specifically, the prompt is:

```
I am in a place1, place2, place3. I see a object1, object2, object3. I am activity1.
Question: What am I doing? Answer: I am most likely
```

The summarization process in general can capture more rich descriptions conditioned on the places, objects, and activities, and qualitatively seem to do well at ignoring irrelevant categories (i.e., denoising). For example:

```
I am in a nursing home, landfill, living room. I see a wine, wine glass, woman. I am
drinking wine. Question: What am I doing? Answer: I am most likely enjoying a glass of
wine with a friend or loved one.
```

However, while the LLM's denoising capabilities can compensate for the shortcomings of the VLM, it is important to note that this may also cause unwanted ignoring of notable, but rare events (e.g., such as witnessing a purple unicorn, which may be ignored, but potentially it is Halloween). Finding new ways in which such events can be indexed appropriately may be useful for downstream applications.

**Egocentric Image Summary Results.** On egocentric images, we show several qualitative examples of summaries generated by our system in Fig. 10, and compare them to results from a state-of-the-art image captioning model, ClipCap (Mokady et al., 2021). While state-of-the-art captioning models can perform reasonably over several of the images, we find that our system generally produces more relevant captions for a larger portion of the egocentric examples. Image captioning models are biased based on the datasets they are trained on, and have shown to perform poorly on egocentric images (Agarwal et al., 2020), which aligns with our observations. Relatively less research has been carried out specifically on egocentric image captioning (Fan et al., 2018). SMs can nevertheless produce reasonable captions without additional training on domain-specific data.

### 3.3-B. Adding Audio into Single-moment Summaries.

In addition to using visual perceptual inputs, we may use a Socratic approach which engages perceptual inputs from audio as well, via an ALM (audio language model). Our example egocentric perception system uses Wav2CLIP (Wu et al., 2021a) as the ALM. Wav2CLIP is trained on 5-second audio clips from the VGGSound dataset (Chen et al., 2020), and is trained in a contrastive manner by aligning its audio encoder to the visual CLIP embeddings from video.

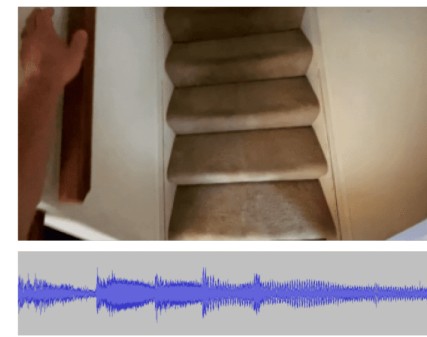

Fig. 12: Example frame and corresponding (centered) 5-second audio clip which provide the driving example for Sec. C.2-B, i.e., adding in ALMs into Socratic dialogue to improve single-moment summarization.

Incorporating an ALM like Wav2CLIP into our Socratic framework can provide an additional modality with which to perform zero-shot cross-modal reasoning, and this may help further improve inference beyond the vision-language-only case. Fig. 12 displays a driving example for which a visual-only summarization produced the less-than-desirable summary: "I am climbing a staircase, and I may see a hamster or human leg" with the incorrect propogation of the false detection of a hamster and human leg.

To perform audio-aided single-moment summarization, we first run image-based summarization as described previously, but we then prompt the LLM to suggest sounds that it may hear, given the visual context, via "⟨visual single-image summary⟩. 5 Possible Sounds:". For the example in Fig. 12 an example prompt, which has already gone through multiple rounds of Socratic dialogue to be generated, together with completion by the LLM is:

```
Places: staircase. Objects: stairs, animal, mammal, hamster, human leg. Activities:
climbing. 5 Possible Sounds: footsteps, creaking stairs, someone calling your name, a
dog barking, a centipede crawling.
```

These auditory entities expressed in language can then be ranked by the ALM. In this moment of the video, the sound of footsteps can be faintly heard in the background, and in this case the ALM provides a correct detection of ranking footsteps as the most likely sound. This ranking can then be incorporated into a prompt for the LLM to provide the single-image summary, for example:

```
I am in a: {place}. I see a: {object1}, {object2}, {object3}, {object4}, {object5}. I
think I hear {sound1} I am: {activity}. Summary: I am most likely
```

As above, incorporating "I hear footsteps" into the summary and prompting this to the LLM provides the completion: "**climbing a staircase, and I may hear footsteps.**" In this case, this summary result is preferable to the mentioned single-image caption without sound.

While this example demonstrates in a certain case the utility of audio-informed summaries, overall in egocentric video, with a variety of background noise, we find that Wav2CLIP can provide reasonable detections for certain language-represented auditory entities such as 'baby babbling' and entities to do with 'running water', but do not provide as robust detections as CLIP. Hence, using an LLM to suggest sound categories conditioned on the visual entities detected by the VLM can improve overall auditory entity accuracy. Also, while there are many advantages to the specific Wav2CLIP approach, including its use of the CLIP embedding space, a major downside is that the training process is "blind" to hearing things that cannot be seen. Accordingly, for the rest of demonstrations shown, we simply build world-state history from VLM-LLM interactions alone. We expect however that with further attention to model approaches, and scaling of audio-language datasets, approaches like Wav2CLIP will increase in robustness. We also show an additional application (Sec. C.3) of audio, for audio retrieval. In that case, only a single auditory search entity is required in order to enable a useful application, and so it can be easier to verify that it is a sufficiently robustly-detected entity.

### 3.3-C. Compiling a Language-Based World-State History

Our system compiles the image summaries from each key video frame into a language-based world-state history. Since the total number of frames in the video may be large, compiling a summary for every individual frame would create text that is too large (too many tokens) to be processed di-

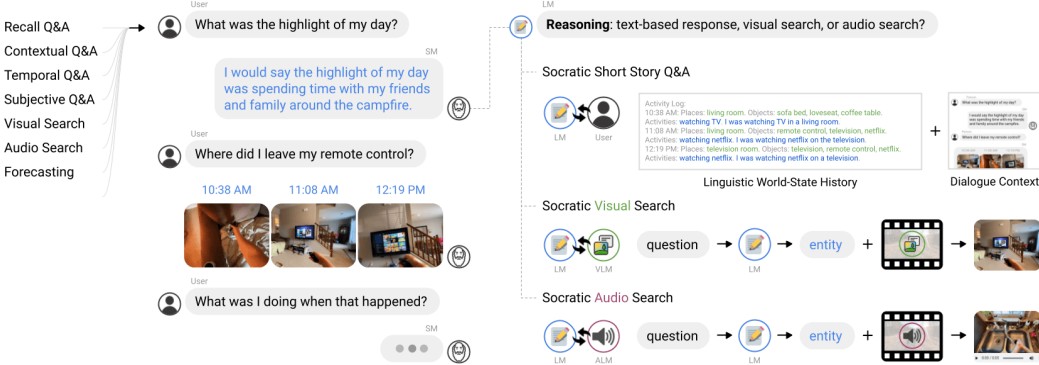

Fig. 13: SMs can interface with the user through dialogue and perform a variety of tasks (formulated as Q&A) with egocentric video: sorting reasoning questions by their output modalities e.g., text-base responses, images from visual search, video snippets from audio search. Depending on the modality, each question can pass through a different sequence of Socratic interactions between the LM, VLM, and ALM.

rectly by an LLM as context for Q&A. Accordingly in this work, we propose solutions that sparsify and/or condense language-based world-state histories (e.g., via search-based methods) into practically usable context sizes for reasoning. In particular, we explore two methods of identifying "key moments" in videos for summarization: (i) uniform sampling over time, and (ii) video search (image or audio retrieval) for on-the-fly compilation of context.

The first method, uniform sampling, is straightforward and compiles a world-state history from Socratic summaries of video frames sampled at fixed time intervals. This can also be condensed hierarchically using recursive linguistic summarization (Wu et al., 2021b), to fit even dense sampling into usable LM-context sizes. However, while broadly indiscriminate, uniform sampling may not have sufficient temporal resolution to capture important spontaneous events in the video (such as adding salt to the pot while cooking soup in the kitchen).

Hence the second method, identifying key moments with video search, uses a VLM or ALM to search for entities most relevant to the question, which can more precisely index the frames in which the subject appears. Specifically, our instantiation of SMs for this component parses a natural language question with an LLM into several search entities to be used to find key frames in the video. For example, the question "did I drink coffee today?" yields a search entity "**drink coffee**" that is then used with language-conditioned video search to index the most relevant $n$ key frames of "drink coffee" in the video. The LLM categorizes the search, which can be image-based (VLMs) or audio-based (ALMs), e.g., for language-conditioned auditory recall questions ((Oncescu et al., 2021)) like "why was my wife laughing today?" . While search-based indexing of key moments can be useful for finding spontaneous events, this method for generating context can also provide disadvantages for downstream Q&A if the answer to the question depends on events that are not directly related to the search subject. For example, "why was I chopping wood today?" returns key frames related to "chopping wood", but does not return the key frames after the event related to making a campfire. On the other hand, if uniform sampling is employed and the campfire events are captured by the summary, then the LLM can successfully return the answer "**I was making a campfire.**" Choosing which method to use for compiling the language-based world-state history may depend on the application.

Drawing analogies to 3D vision and robotics, the world-state history can be thought of as building an on-the-fly reconstruction of events in the observable world with language, rather than other representations, such as dynamically-updated 3D meshes (Izadi et al., 2011) or neural fields (Tancik et al., 2022).

**Language-based World-state History Results.** Fig. 10, middle, shows results generated by our system. The specific event log shown in Fig. 10 has been trimmed down for space considerations, but is representative of the type of event logs that may be generated without manual curation. These event logs are used as context to enable LLM open-ended reasoning on video, as demonstrated in the next section.

### C.3 OPEN-ENDED REASONING ON EGOCENTRIC VIDEO

In this section we describe a few examples of how the Socratic Models framework can be used to perform open-ended multimodal-informed completion of text prompts, conditioned on egocentric video (examples in Fig. 9). There are of course limitations to what they can provide, but our demonstrated examples suggest that we can already today generate compelling answers to open-ended reasoning tasks, at a scope that is beyond what we are aware is possible today with available methods. Of course, the answers may also inherit undesirable characteristics from the component models, such as an LLM that is overconfident even when wrong. It is our hope that our results may help inspire work on preparing even more comprehensive video understanding datasets for the community, to assist further assessment.

Our example system uses a language-based world-state history generated through Socratic multi-model discussion (Sec. C.2), and provides this as context to an LLM to enable open-ended reasoning on egocentric videos. Open-ended text prompts from a user, conditioned on an egocentric video, can yields three types of responses: a text-based response, a visual result, and/or an audio clip. These latter two provide examples that open up the capabilities of the system to respond not only with text-based responses, but also respond with video snippets themselves, which may be a higher-bandwidth way to respond to user requests (*"a picture is worth a thousand words"*). The specific composition of our system is of course just one example – overall, the modularity of the Socratic approach makes it easy to compose together foundation models, zero-shot, in a variety of ways to provide a spectrum of multimodal reasoning capabilities.

The demonstrated tasks include (i) summarization, (ii) open-ended Q&A, (iii) forecasting, (iv) corrections, and (v) video search for either visual or audio cues. These tasks have predominantly been studied in isolation in the research community – but our example results with SMs suggest they can be subsumed under the same unified language-based system for multimodal reasoning.

**(i) Summarization** can be implemented by prompting an LLM to complete the excerpt "{world-state history} Summary of my day:" to which it can respond with outputs like "**I slept in a bed, made coffee, watched TV, did laundry, received a package, bench pressed, showered, ate a sandwich, worked on a computer, and drank wine.**" Since the language-based world-state history is constructed with summaries of visual content, it carries contextual information that can be complementary to what is found in closed captions (e.g., speech and dialogue). Summarizing egocentric videos enables a number of applications, including augmenting human memory to recall events, or life-logging of daily activities for caregiver assistance. Our system draws similarity to early work in the area involving text-based summarization and identifying key frames (see (Barbieri et al., 2003) for an early survey and (Del Molino et al., 2016; Apostolidis et al., 2021) for more recent surveys).

**(ii) Open-ended Q&A** can be implemented by prompting the LLM to complete the template: "{world-state history} Q: {question} A:". We find that LLMs such as GPT-3 can generate surprisingly meaningful results to binary yes or no questions, contextual reasoning questions, as well as temporal reasoning questions. As in (Yang et al., 2021) we can further prompt the LLM to *explain the answer* by adding "This is because:". We find that the accuracy of the answers and explanations remain largely conditioned on whether the necessary information can be found within the world-state history. This suggests that the quality of the language-based reconstructions of the videos (e.g., via key frame sampling and captioning in this work) is central to the approach.

We show qualitative examples of free-form question answering using our SM system on egocentric video in Fig. 10, bottom, Fig. 11, and Fig. 13 generated using a first-person POV video[4] as input.

***Recall Questions.*** SMs can perform simple retrieval of events. For example, "did I eat dinner today?", yields a response "**yes I ate dinner today.**" along with an explanation "**I was seen eating a sandwich in a kitchen at 5:27 PM.**" which points to the key frame that was captioned with the sandwich in hand. Another example that involves contextual reasoning to recall events is "what was I doing outdoors?" to which the system responds "**I was chopping wood in a yard.**" Likewise, if the entities described in the question do not appear in the world-state history, such as "did I drive today?" the system can respond with a negative answer: "**no, I did not drive today.**" with an explanation "**I was at home all day.**" This capability expands beyond standard video search, which

---

[4]Examples on https://youtu.be/-UXKmqBPk1w used with permission from Cody Wanner.

might only return nearest neighbor video frames, without a natural language response (or a negative response).

The performance of recalling events largely depends on the relevance of the language-based world-state history to the question. We find that recall-type questions work best with world-state history logs that are compiled by using search-based key frame indexing (see Sec. 3.3-B). The system can still return negative responses, since the captioning of the key frames are not influenced by the question.

***Temporal Reasoning.*** SMs can answer questions related to time by appending timestamps to each key moment in the world-state history. By associating image summaries to times of the day, this allows answering questions that time index various activities. For example "when did I last drink coffee?" can return the last time drinking coffee was mentioned in the log, with a full response "**I last drank coffee at 10:17 AM**" and an explanation "**I was making coffee in the kitchen.**" The system can also count events, for example when asked "how many times did I receive a package today?", the system will respond appropriately "**I received a package once today.**" with an explanation "**I was receiving a package at 3:24 PM**". We find that a common failure mode for these types of questions is that the system tends to over-count, especially as a reaction to false positive VLM detection results that get surfaced into the world-state history. For example, asking "who did I interact with?" would yield "**woman, hamster**" where hamster was a false positive prediction from CLIP. These issues become more prominent with search-based key frame sampling, as a byproduct of an inability to distinguish neighboring local argmaxes of the same event from each other.

***Cause and Effect Reasoning.*** SMs can answer questions about cause and effect relationships between events, conditioned on that all the events appear in the world-state history. For example, when asked "why did I go to the front porch today?" the system would respond "**I went to the front porch today to receive a package.**" and an explanation "**I saw on the porch a package and knew that I was expecting it.**" These types of questions are exciting because they suggest opportunities for prompting logical deduction of events. However, since information about both the cause and the effect needs to be in the world-state history, the quality of results remains highly dependent on the key frame sampling strategy used to compile it (Sec. 3.3-B). Uniform gives an unbiased account of events, and is currently the best variant for this form of reasoning. More targeted construction of the world-state history with search based key frames can sometimes miss frames that capture the answer to the question.

***Subjective Reasoning.*** SMs can also answer more subjective questions, such as "was I happy today?" or "what was my favorite drink today?". Without additional context, these questions rely on biases from the LM's dataset – which could have negative consequences, and should be managed carefully with additional mechanisms for safety and groundedness (Thoppilan et al., 2022). The full personalization of these subjective questions are likely to be conditioned on whether a better context can be constructed of prior user behaviors related to the question.

**(iii) Forecasting** of future events can be formulated as language-based world-state completion. Our system prompts the LLM to complete the rest of an input event log. Timestamps of predictions can be preemptively specified depending on application needs. The completion results are generative, and more broad than binary event classification (e.g., (Lei et al., 2020)). Example completion (also shown in Fig. 9):

```
1:46 PM: I am eating a sandwich in a kitchen.
2:18 PM: I am checking time and working on a laptop in a clean room.
2:49 PM: I am buying produce from a grocery store or market.
3:21 PM: I am driving a car.
4:03 PM: I am in a park and see a playground.
4:35 PM: I am in a home and see a television.
```

Few-shot prompting the LLM with additional examples of prior event logs most similar to the current one is likely to improve the accuracy of the completion results. Without additional context, these results are again biased towards typical schedules seen by the LLM across Internet-scale data.

To a certain extent, this forecasting capability extends and generalizes the traditional topic of activity forecasting in computer vision. In the research community, activity forecasting has been often formulated as an extension of action classification, tracking, or feature generation: Given a sequence of image frames, they directly predict a few categorized actions (Ryoo, 2011; Hoai & De la Torre,

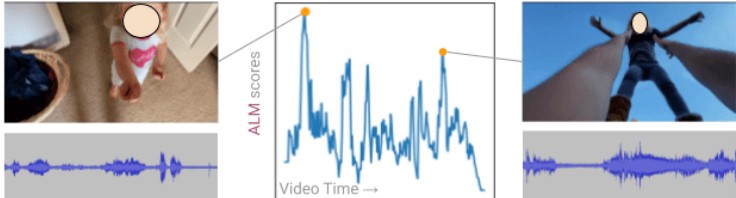

Fig. 14: Example zero-shot language-prompted auditory retrieval (shown: top 2 results) in response to "what did my daughter's laugh sound like today?", for which an LLM identifies the audio search query of "daughter's laugh", and an ALM (Wav2CLIP) is used for audio retrieval. The top (left) retrieval is only partially correct, returning a video clip involving the daughter but not laughter. The second (right) retrieval is correct, from a moment of playing (getting tossed into the air). Faces obscured for privacy.

2014; Rhinehart & Kitani, 2017), human locations (Kitani et al., 2012), or image features (Vondrick et al., 2016) to be observed in the future frames. In contrast, Socratic Models with LLMs enables generating more semantically interpretable descriptions of future events, conditioned on multimodal information.

**(iv) Corrections.** SMs can be prompted to incorporate human feedback in the loop as well, which could be useful for interactive language-based systems. For example, given image captions generated from an VLM and LM:

```
Context: Where am I? outdoor cabin, campsite, outdoor inn.  What do I see?  fire,
marshmallow, fire iron, hearth, fireside, camp chair.  What am I doing?  Commonsense
suggests: roasting marshmallows, sitting around the fire, chatting. Most likely: sitting
around the fire.
Original Summary: I am camping and enjoying the company of my friends around the fire.
Corrections: It was actually my family, not friends, sitting around the fire.
Corrected Summary: I am camping with my family and enjoying the company of them around
the fire.
```

**(v) Video Search: Image or Audio Retrieval.** Our SM system can also return additional modalities (images, audio) as answers to questions, by simply few-shot prompting the LLM to classify a target modality based on the input question. For example, "where did I leave my remote control" can map to image search using VLM features for "remote control" while "what did my daughter's laugh sound like today?" can map to natural-langauge-queried audio search ((Oncescu et al., 2021)) using ALM features for "daughter's laugh" (Fig. 14). This can be useful for some applications (e.g., AR) in which the user may find the retrieved modality to be more useful than a natural language response. Our approach for this uses an LLM to parse a search entity from the question to index key video frames. This is done with several few-shot examples provided as context. For example, the question "when did I last wash my hands?" yields a search entity "**wash my hands**" that is then used with video search to index the most relevant $n$ key frames of "wash my hands" in the video. Specifically, our system runs video search by ranking matching CLIP or Wav2CLIP features of the entity text against all video frames, and returning the top $n$ local maximums. For each frame, the features can either be image features or audio features (e.g., from the surrounding 5 seconds with Wav2CLIP) – where the LLM few-shot categorizes which domain to use for any given question. This can be thought of as calling different subprograms for hierarchical search.

**Limitations.** Overall, our results suggest that SMs are capable of generating meaningful outputs for various egocentric perception tasks via visual contextual reasoning – but its limitations also suggest areas for future work. One primary bottleneck in the Q&A system is that it relies on the richness (i.e., recall) and quality (i.e., precision) of the event log. For example, on "counting" questions: if the construction of a world-state history produces many false positive predictions of e.g. laptops, televisions, tablets, etc. then asking the language model to count how many times electronics were seen during the video could be inaccurate. This likely could be improved with better visual and audio detectors, or captioning systems (Gu et al., 2021). Also, we find that the used Wav2CLIP may provide satisfactory results for certain categories in audio retrieval, but we currently do not involve it in generating the event log, since its robustness and range of open-language detection is not at the same level as CLIP. This seems addressable with further approaches and scaling of datasets in the audio-language domain.

Additionally, accurate response to cause and effect reasoning questions also require relevant key moments to be reflected in the event log – which points to open ended questions on how to achieve

better key frame sampling (beyond the simple baselines that we have demonstrated). Finally, the dialogue between the different models are fairly structured with manually engineered prompts. It may be interesting to investigate more autonomous means of achieving language-based closed loop discussions between the models until a commonsense consensus is reached.

## D    SCALING UP SOCRATIC VIDEO SEARCH

The search algorithms of the SMs, which may be used both for compiling world-state history (Sec. C.2-C) and for video search retrieval (Sec. C.3) rely on the matching procedure conducted in the corresponding latent space (e.g., VLM features of the text snippet against these of the video frames). This can be abstracted as dot-product-maximization key search in the given key-dataset. In practice, if the key-dataset is large (e.g., long videos) a naive linear search is prohibitively expensive. We propose several solutions to this problem.

**MIP-Search.**    The first observation is that several data pre-processing techniques applied in the so-called *maximum inner product* (MIP) search can be directly used to reorganize the keys (e.g., latent representations of video frames) to provide sub-linear querying mechanism for the incoming text snippet (see: (Abuzaid et al., 2019)). Those include pruning and various indexing techniques, such as LSH-hashing (Shrivastava & Li, 2014). In the hashing approach, a collection of hash-tables, indexed by the binarized representations of the hashes is stored with different entries of the hash table corresponding to the subsets of keys producing a particular hash. There are several cheap ways of computing such hashes, e.g., *signed random projection* (those in principle linearize the angular distance, but every MIP task can be translated to the minimum angular distance search problem). The querying is then conducted by searching for the most similar hash-entries in the hash-tables and then performing linear search only on the subsets of keys corresponding to these entries to obtain final ranking.

**Associative Memories.**    The above approach provides sub-linear querying mechanism, but does not address the space complexity problem. In the scenario of strict memory requirements, we propose to leverage recently introduced techniques on linear attention (Choromanski et al., 2021b) combined with *modern continuous associative memory* (MCAM) models (Ramsauer et al., 2021). MCAM models are de facto differentiable dictionaries (with provable few-shot retrieval) that can be thought of as energy-based models using negated exponentiated latent-representations-dot-product energy for the *exponential* storage capacity. A naive computation of such an energy still requires explicitly keeping all the patterns (which is exactly what we want to avoid), but this can be bypassed by applying the linearization of that energy (which effectively is just the negated sum of the softmax kernel values) with the FAVOR+ mechanism used in linear-attention Transformers, called *Performers* (Choromanski et al., 2021b). This modification has several advantages: (1) it makes the size of the dictionary completely independent from the number of the implicitly stored patterns; the size now scales linearly with the number of random features used for energy linearization, (2) it provides a *constant-time* querying mechanism at the price of compressing all the patterns (and thus losing some information).

**Random Feature Trees.**    The other approach, that combined the ideas from both MIP-search and linear attention systems, leverages the so-called *random feature tree* (RFT) data structure (Rawat et al., 2019). This approach relaxes the MIP-search to sampling from the linearized softmax distribution via FAVOR+ (Choromanski et al., 2021a). Sampling from such a linearized distribution can be done in time logarithmic in the number of samples via RFT which is a balanced tree with leaves corresponding to latent representations of video frames and nodes encoding representations of the subsets of keys (e.g., the video frames) defined as sums of the random feature transforms of the keys.

## E    MULTIMODAL ASSISTIVE DIALOGUE: FULL PROMPT AND OUTPUTS

The approach preconditions an LLM with context that includes when and how key phrases should be referenced:

```
Alice is a an expert chef that will help Bob prepare a given recipe. If Bob asks for the next step, Alice will
respond with "Step: " followed by the next step of the recipe. If Bob does not have the right ingredients, Alice
will assist Bob in finding suitable replacements. If Bob asks Alice to describe something that is better shown
visually, Alice will say "(image)" followed by a response. Ingredients: {ingredients} Recipe: {recipe}
```

```
Alice: Step: In a medium bowl, beat the egg whites until frothy.
Bob: how many eggs.
Alice: 3 egg whites, room temperature
Bob: whats room temperature.
Alice: Room temperature is around 70 degrees Fahrenheit.
Bob: what do you mean by medium bowl.
Alice: A medium bowl is a bowl that is big enough to fit the ingredients for this
recipe and that is comfortable to mix in.
Bob: exactly how big should that be.
Alice: (image) A medium bowl is a bowl that is big enough to fit the ingredients for
this recipe and that is comfortable to mix in.
(a) video search {recipe video} with "A medium bowl is a bowl that is..."
Bob: ok assume im done whats next.
Alice: Step: Keep beating and slowly add the sugar until stiff peaks form.
Bob: whats stiff peaks.
Alice: Stiff peaks are when the egg whites stand up in peaks that do not fall over.
Bob: can you show me what that looks like.
Alice: (image) Stiff peaks are when the egg whites stand up in peaks that do not fall
over.
(b) video search {recipe video} with "Stiff peaks are when the egg whites..."
Bob: do i need to add water.
Alice: No, you do not need to add water.
```

Fig. 15: Multimodal prompt engineering VLM, Web Search, and LM can enable multimodal dialogue applications such as guiding a user through online recipe steps and providing assistive visuals via video search.

## F    ADDITIONAL NOTES ON ROBOT EXPERIMENTS

Full robot demonstration from the main paper:

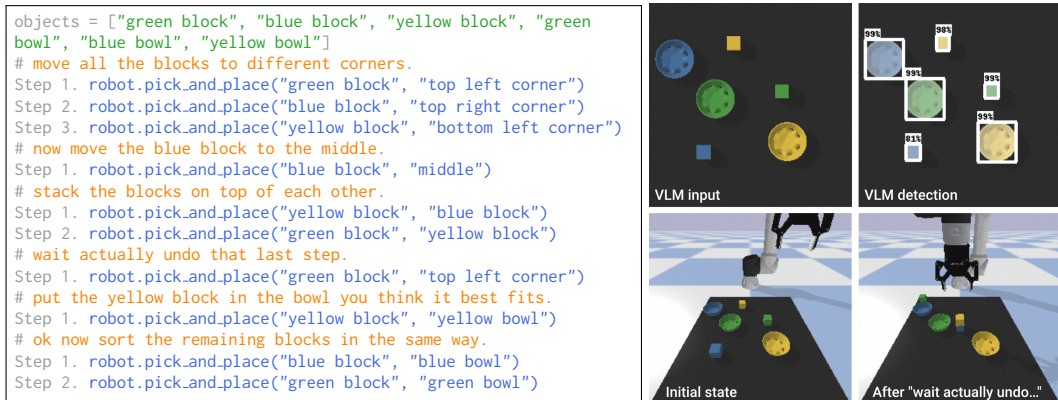

```
objects = ["green block", "blue block", "yellow block", "green
bowl", "blue bowl", "yellow bowl"]
# move all the blocks to different corners.
Step 1. robot.pick_and_place("green block", "top left corner")
Step 2. robot.pick_and_place("blue block", "top right corner")
Step 3. robot.pick_and_place("yellow block", "bottom left corner")
# now move the blue block to the middle.
Step 1. robot.pick_and_place("blue block", "middle")
# stack the blocks on top of each other.
Step 1. robot.pick_and_place("yellow block", "blue block")
Step 2. robot.pick_and_place("green block", "yellow block")
# wait actually undo that last step.
Step 1. robot.pick_and_place("green block", "top left corner")
# put the yellow block in the bowl you think it best fits.
Step 1. robot.pick_and_place("yellow block", "yellow bowl")
# ok now sort the remaining blocks in the same way.
Step 1. robot.pick_and_place("blue block", "blue bowl")
Step 2. robot.pick_and_place("green block", "green bowl")
```

Fig. 16: Multimodal prompt engineering VLM, LM, language-conditioned policies (e.g., via CLIPort (Shridhar et al., 2022)) can enable robots to parse and generate plans from free-form human instructions (in orange).

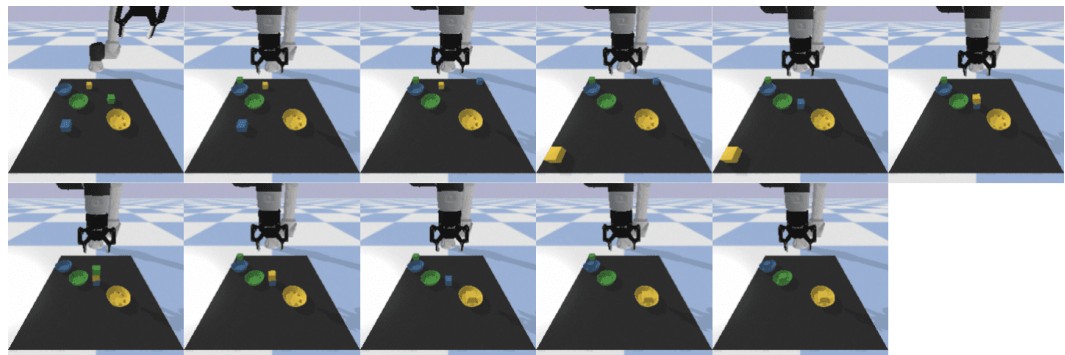

Fig. 17: Full rollout of the robot environment for the example presented in Sec. 5.3 of the main paper.

The SM robot system uses a VLM (open-vocabulary object detection with ViLD (Gu et al., 2021)) to describe the objects in the scene, feeds that description as context to a LLM as a multi-step planner (Ahn et al., 2022; Huang et al., 2022), that then takes as input a natural language instruction and generates the individual steps to be passed to a pretrained language-conditioned robot policy, for which we specifically use a CLIP-conditioned (Shridhar et al., 2022) No-Transport baseline from Zeng et al. (Zeng et al., 2020) (inspired by CLIPort (Shridhar et al., 2022) for open vocabulary pick-and-place). The full prompt used as context to the LLM for multi-step planning is:

```
objects = ["cyan block", "yellow block", "brown block", "green bowl"]
# move all the blocks to the top left corner.
Step 1. robot.pick_and_place("brown block", "top left corner")
Step 2. robot.pick_and_place("cyan block", "top left corner")
Step 3. robot.pick_and_place("yellow block", "top left corner")
# put the yellow one the green thing.
Step 1. robot.pick_and_place("yellow block", "green bowl")
# undo that.
Step 1. robot.pick_and_place("yellow block", "top left corner")
objects = ["pink block", "gray block", "orange block"]
# move the pinkish colored block on the bottom side.
Step 1. robot.pick_and_place("pink block", "bottom side")
objects = ["orange block", "purple bowl", "cyan block", "brown bowl", "pink block"]
# stack the blocks.
Step 1. robot.pick_and_place("pink block", "orange block")
Step 2. robot.pick_and_place("cyan block", "pink block")
# unstack that.
Step 1. robot.pick_and_place("cyan block", "bottom left")
Step 2. robot.pick_and_place("pink block", "left side")
objects = ["red block", "brown block", "purple bowl", "gray bowl", "brown bowl", "pink block", "purple block"]
# group the brown objects together.
Step 1. robot.pick_and_place("brown block", "brown bowl")
objects = ["orange bowl", "red block", "orange block", "red bowl", "purple bowl", "purple block"]
# sort all the blocks into their matching color bowls.
Step 1. robot.pick_and_place("orange block", "orange bowl")
Step 2. robot.pick_and_place("red block", "red bowl")
Step 3. robot.pick_and_place("purple block", "purple bowl")
```

Fig. 17 depicts a full rollout of the example in Sec. 5.3 in the main paper, which involves human dialogue. Fig. 18 shows additional examples of multi-step tasks that the system can perform out-of-the-box with zero-shot SMs. The system is able to reason over order and nuanced language (clockwise vs. counterclockwise) as well as respond to different objects being detected (the stacking task with varied block colors). Note that the LLM is few-shot prompted to generate pseudo code "robot.pick_and_place("A", "B")" which calls a function to return a fixed template sentence "Pick the A and place it on the B." subsequently fed as input to the language-conditioned robot policy. While we can prompt the LLM to directly produce the template sentences as opposed to code, we find that the LLM can sometimes generate phrases or prepositions that are beyond the training set of the language-conditioned policy. We observe that the policies are more likely to return correct actions when the templates can be engineered to be more similar to the phrases seen within the policy's training data. We also found ViLD and CLIP to be brittle in this scene, as the scene is simulated and the objects are not natural. High-performance in this setting requires a good view angle (an overhead camera), filtered colors (red, green, yellow, and blue), and tuned names (we referred to the blocks as "boxes" and the bowls as "circles" to account for the overhead view). Without these changes, the system still is able to complete many tasks, but less consistently. We expect off-the-shelf VLMs of the future to be more robust than those currently available.

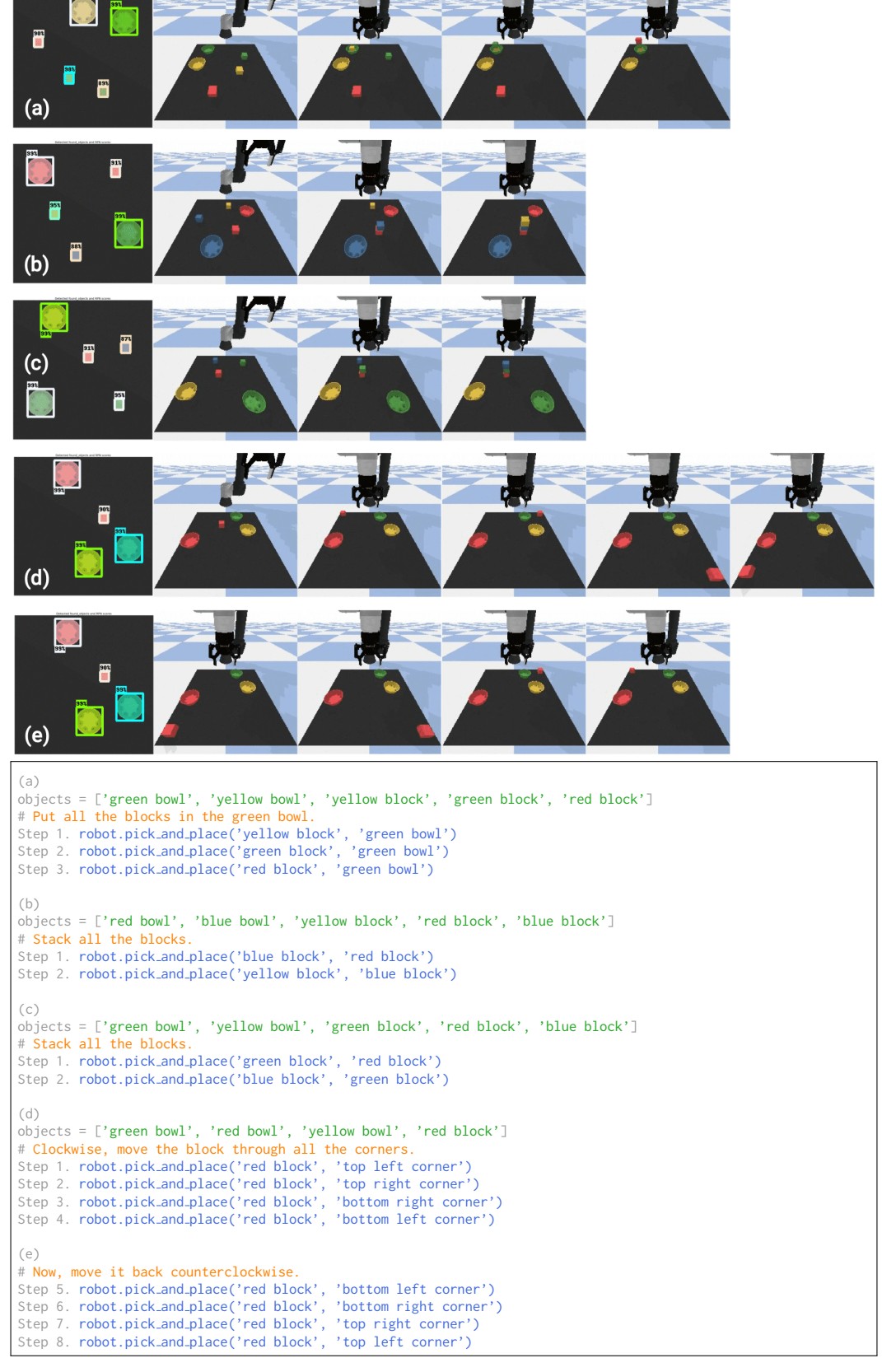

```
(a)
objects = ['green bowl', 'yellow bowl', 'yellow block', 'green block', 'red block']
# Put all the blocks in the green bowl.
Step 1. robot.pick_and_place('yellow block', 'green bowl')
Step 2. robot.pick_and_place('green block', 'green bowl')
Step 3. robot.pick_and_place('red block', 'green bowl')

(b)
objects = ['red bowl', 'blue bowl', 'yellow block', 'red block', 'blue block']
# Stack all the blocks.
Step 1. robot.pick_and_place('blue block', 'red block')
Step 2. robot.pick_and_place('yellow block', 'blue block')

(c)
objects = ['green bowl', 'yellow bowl', 'green block', 'red block', 'blue block']
# Stack all the blocks.
Step 1. robot.pick_and_place('green block', 'red block')
Step 2. robot.pick_and_place('blue block', 'green block')

(d)
objects = ['green bowl', 'red bowl', 'yellow bowl', 'red block']
# Clockwise, move the block through all the corners.
Step 1. robot.pick_and_place('red block', 'top left corner')
Step 2. robot.pick_and_place('red block', 'top right corner')
Step 3. robot.pick_and_place('red block', 'bottom right corner')
Step 4. robot.pick_and_place('red block', 'bottom left corner')

(e)
# Now, move it back counterclockwise.
Step 5. robot.pick_and_place('red block', 'bottom left corner')
Step 6. robot.pick_and_place('red block', 'bottom right corner')
Step 7. robot.pick_and_place('red block', 'top right corner')
Step 8. robot.pick_and_place('red block', 'top left corner')
```

Fig. 18: Additional examples of multi-step tasks that the SM robot system can perform out-of-the-box.

# G SOCRATIC DEDUCTIVE REASONING

In the context of egocentric perception, we find that formulating video Q&A as reading comprehension in SMs directly leverages the extent to which large LLMs are capable of logical reasoning by connecting commonsense relationships with knowledge learned from Internet-scale data. For example, the system returns the following answer when presented with the world-state history log:

```
8:00 AM: went to grocery store to buy orange juice, chocolate, and bread.
8:15 AM: I went to gas station to fill up the vehicle tank.
8:30 AM: drove back home and left the groceries in the kitchen.
8:45 AM: started cooking eggs in the pan.
9:00 AM: the dog went into the kitchen.
9:15 AM: took the dog out for a walk.
9:30 AM: the dog is sick.
Q: Why is the dog sick? A: The dog may have eaten something it was not supposed to,
such as chocolate.
```

Arriving at the answer requires bridging multiple connections between observations e.g., the dog went into the kitchen, the groceries are still in the kitchen, and the groceries contain chocolate. Such results offer a glimpse of what might be possible using SMs for deductive reasoning across multiple domains of information, and raises interesting research questions on (i) how to better assemble language-based world-state histories (beyond what is presented in this work) that capture relevant evidence to improve the accuracy of conclusions, and (ii) how to elicit chain of thought prompting (Wei et al., 2022) to decompose multi-step problems into intermediate ones. For example, one promising extension could be prompting the LLM with chain of thought sequences to expand on hypotheses:

```
Q: What are reasons for why I might be chopping wood? A: Reasons might include: needing
firewood, wanting to make a statement, or needing the exercise.
```

to which each hypothesis can be progressively explored by downstream subprograms called at recursively higher resolutions until a conclusion is reached. These directions suggest pathways towards achieving increasingly meaningful utility and analysis by digital multimodal assistants.

# H BROADER IMPACTS

SMs encourage the use of language as the middleware to build AI systems using off-the-shelf large pretrained models without additional data collection or model finetuning. This leads to several practical benefits, new applications, and risks as well. For one, SMs provide an interpretable window, through language, into the behavior of the systems (even for non-experts). Further, the barrier of entry for this technology is small: SMs can be engineered to capture new functionalities with minimal compute resources, and to tackle applications that have traditionally been data-scarce. No model training was used to create our demonstrated results. This can be enabling, but also raises potential risks, since it increases the flexibility of unintended end use applications, and should be carefully monitored over time. It is also important to note that the system may generate results that reflect unwanted biases found in the Internet-scale data on which incorporated models are trained, and should be used with caution (and checked for correctness) in downstream applications. We welcome broad discussion on how to maximize potential positive impacts (enabling broad, new multimodal applications, with minimal resources) while minimizing the capabilities of bad actors.

**Energy and Resource Consumption** Regarding the impact on energy and other resource consumption, this work may help pave a path for new, capable machine learning models to be composed with minimal training resource consumption, provided that large foundational pretrained models are available. This may help provide an answer for how large pretrained models may be retargeted to a wide variety of multimodal applications, without additional considerable compute resources required. Since SMs help demonstrate how a wide variety of applications may be addressed with fixed (pretrained) models zero-shot, this may also help foster adoption of new machine learning accelerators (e.g., fixed analog circuity (Reuther et al., 2020), optical diffraction (Lin et al., 2018)) for inference with substantially lower power consumption and more compact form factors.

