# OpenReview forum: "Socratic Models: Composing Zero-Shot Multimodal Reasoning with Language"
_ICLR.cc/2023/Conference — ICLR 2023 notable top 25%_

### Official Review · Reviewer_NinX · 2022-10-22

**Confidence:** 4
**Clarity, Quality, Novelty And Reproducibility:** see above
**Correctness:** 4
**Technical Novelty And Significance:** 3
**Empirical Novelty And Significance:** 3
**Recommendation:** 8

**Strength And Weaknesses:**

My favorite part of this work is the creativity of the compositions.
For captioning, the authors compose several types of object detectors
(scene type, person counter, object detector, etc.), format the
outputs of that in a LLM prompt, which proposes several captions,
which are then re-ranked by CLIP: the resulting model is SOTA for
zero-shot captioning, and even competitive with models that fine-tune
on upaired captions. The rest of the examples are even more creative,
ranging from video retrieval to egocentric video summarization, to
multimodal assistive dialogue, and beyond.

Other positives:

- I think these things make for great baselines, like the authors
  said. I think most papers on multimodal modeling should include one.

- I like that the features passed between models are in text format,
  i.e., they are more interpretable compared to visual features from
  an vision model directly.

- I like that the decoupled nature of these compositions means that
  performance improvements outside of multimodal modeling (e.g.,
  better language-only models or better vision-only models) translate
  directly to improvements in socratic models.

My biggest technical concerns are:

- The granularity of the outputs of the vision/audio models, when put
  into text, may not be sufficient for every downstream task ---
  because information is lost, e.g., when making an object
  classification vs. the whole image, socratic models may hallucinate
  moreso than other approaches. The authors are aware of this, and
  it's interesting to think about how to fix this.

- The design of the prompts, compositions, etc. seem quite ad-hoc,
  i.e., it seems like there's probably a lot of "art" to creating a
  good prompt/set of attributes to textify from a VLM, etc.

My biggest presentation concern is:

- Are we really going to call any composition of vision/audio/language
  model a "socratic" model? Even ones that don't have as much
  "discussion"? It seems like the models are not really engaging with
  a "dialogue" as one might imagine given the name of the method.

**Summary Of The Paper:**

The authors propose composing multimodal models with language models
in a process they call "socratic models". The key idea is that the
outputs of VLMs/ALMs can be reformulated in text, which can then be
fed to a LLM via a prompt. While each instantiation of a socratic
model is slightly different, this family of approaches can serve as a
surprisingly strong zero-shot baseline for multimodal tasks. The
authors consider several creative compositions and demonstrate that
their general approach is more effective than prior efforts for zero
and few shot multimodal models.

**Summary Of The Review:**

Overall, this is a creative paper with some very promising results.  I
am hopeful that future multimodal works will include a socratic model
at least as a baseline. And --- I think the future directions for
these types of models (i.e., models that communicate via textual
representations) are quite promising --- I can't wait to see where it
goes!

---

> ### Author Response · Authors · 2022-11-17
> **Response to Reviewer NinX**
>
> We thank the reviewer for their encouraging comments, and highlighting the positives of our approach, including that their *"favorite part of this work is the creativity of the compositions"*. We've made some valuable additions to our work based on all the reviewers' feedback, as well as a few responses below to some of the comments from the reviewer.
>
> > *"I think these things make for great baselines, like the authors said."*
>
> We agree. In domains where there exists abundant data, it is likely to be preferred to finetune a model there, but these models can serve as reasonable zero-shot baselines.
>
> > *"I like that the decoupled nature of these compositions means that performance improvements outside of multimodal modeling (e.g., better language-only models or better vision-only models) translate directly to improvements in socratic models."*
>
> We agree, and some good recent evidence for this is that, while running the additional requested ablations for Reviewer J1Ki, it turns out that our image captioning implementation improved over the past couple months, just due to GPT-3 improving behind the public API.
>
> > *"The design of the prompts, compositions, etc. seem quite ad-hoc, i.e., it seems like there's probably a lot of "art" to creating a good prompt/set of attributes to textify from a VLM, etc."*
>
> The reviewer may be interested in our responses to Reviewer J1Ki, where we added additional experiments per their requests to try different types of prompt wording / other attributes ablations.
>
> > *"Are we really going to call any composition of vision/audio/language model a "socratic" model?... It seems like the models are not really engaging with a "dialogue" as one might imagine given the name of the method."*
>
> Yes it’s a good question, we hope the community can decide over time if it is a useful term or not. We thought it was at least useful for the scope of our paper to have a term for this class of systems, and yes the name is meant to be “loosely inspired by” (see start of Section 3) rather than a literal analogy.

---

> > ### Comment · Reviewer_NinX · 2022-11-29
> > **ack**
> >
> > just acknowledging that I read through your response and will take it into account during additional reviewer discussion. I remain positive in my evaluation.

---

### Official Review · Reviewer_x3bb · 2022-10-24

**Confidence:** 3
**Correctness:** 3
**Technical Novelty And Significance:** 3
**Empirical Novelty And Significance:** 3
**Recommendation:** 6

**Clarity, Quality, Novelty And Reproducibility:**

This paper is well-written, the proposed idea is original. The proposed method is novel and well-motivated and this paper provides evaluations on multiple tasks, but lack enough statistical evaluations in many zero-shot applications (Section 5).

**Strength And Weaknesses:**

Strengths
1. Interesting methods by effectively incorporating LMs from different modalities
2. Strong results on zero/few-shot learning
3. Demonstrate interesting examples on many zero-shot applications

Weaknesses
1. Combining different LMs would make the inference speed much slower, which could be an obstacle for the application.
2. Section 5 only provides several examples without statistical evaluation results. It it hard to evaluate the model's performance in this application, since the examples could be cherry pick.
3. In the image captioning evaluation (MS COCO) in Table 1, there is still a large gap between SMs and ClipCap. More few-shot baselines are needed for evaluating the image captioning quality of SMs.

**Summary Of The Paper:**

This paper studies how to incorporate vision language model (VLM), audio language model (ALM) and large language model (LLM) to perform joint predictions for multimodal tasks. The integration of different LMs are based on various language prompts. Results show that the proposed method achieves promising zero-shot and few-shot performance. In addition, the proposed method can be applied to many novel zero-shot multimodal applications.

**Summary Of The Review:**

This paper proposed Socratic Models which aim to incorporate VLM, ALM and LLM for zero-shot multimodal tasks. The proposed idea is well motivated and novel and the experiments show promising and strong zero-shot results. However, this method might slow down the inference speed, and this paper lacks enough statistical evaluations in many zero-shot applications.

---

> ### Author Response · Authors · 2022-11-17
> **Response to Reviewer x3bb**
>
> We thank the reviewer for their feedback. We are glad that they think the work *"is novel and well-motivated"*, and we also appreciate their other comments and suggestions for which we have responses below.
>
> > *"Combining different LMs would make the inference speed much slower, which could be an obstacle for the application."*
>
> Agreed – using multiple (large) LMs may cause inference time to increase, which could impede specific applications that are time-sensitive. However, since the LM (or other large) pre-trained component models can be frozen and repurposed for several different applications as demonstrated, this potentially points to an interesting future with the next generation of accelerated machine learning models, such as fixed analog circuitry **[1]** or optical diffraction **[2]**. These could provide inference with substantially lower power consumption, faster speeds, and more compact form factors (see Appendix Section H for a brief discussion).
>
> Additionally, beyond accelerating a specific model, since SMs use language as a standardized interface between models, this makes it possible to swap in smaller and faster models without additional training, for applications that have certain inference speed requirements.
>
>
> > *"Section 5 only provides several examples without statistical evaluation results. It it hard to evaluate the model's performance in this application, since the examples could be cherry pick."*
>
> Yes, our goals are to (i) in Section 4 provide quantitative evaluations for selected applications where there exist strong benchmarks and pre-existing (zero- and few-shot) baselines, and then (ii) in Section 5 highlight a few application domains that are scarce in data and not well-covered by benchmarks, but for which we can assemble a zero-shot system.  Our hope is that such zero-shot baselines could be considered in future work when benchmarks may be created that address .e.g., long-form egocentric video understanding, or open-ended dialogue with manipulation robots.  Further, we have provided open-source implementations available to experiment with and get a broad sense of capabilities.
>
> Please also, if interested, see our response to Reviewer J1Ki, where we added considerably more quantitative evaluations overall to the paper per their suggestions, including trying different wordings of prompts, ablating categories of information, and several others.
>
> > *"In the image captioning evaluation (MS COCO) in Table 1, there is still a large gap between SMs and ClipCap. More few-shot baselines are needed for evaluating the image captioning quality of SMs."*
>
> In domains that are data rich (Section 4), we agree that the zero- and few-shot results are still far from those of methods finetuned on the training set – for the moment, they can only serve as baselines. Under this premise, we shaped our experiments to primarily focus on comparisons to other published zero-shot methods.
>
> ---
>
> **[1]** *Albert Reuther et al. "Survey of Machine Learning Accelerators." IEEE-HPEC, 2020.*
>
> **[2]** *Xing Lin et al. "All-optical machine learning using diffractive deep neural networks." Science, 2018.*

---

### Official Review · Reviewer_J1Ki · 2022-10-24

**Confidence:** 4
**Correctness:** 3
**Technical Novelty And Significance:** 3
**Empirical Novelty And Significance:** 3
**Recommendation:** 8

**Clarity, Quality, Novelty And Reproducibility:**

Please see strengths and weaknesses for more details. Availability of corresponding code ideally makes the work reproducible while there is limited technical novelty. The paper is written clearly in most parts.

**Strength And Weaknesses:**

Strength:
- This paper is well-motivated where zero-shot multimodal reasoning is an important research area with vast industrial and research applications.
- The authors promise the availability of corresponding code and also provide anonymous colab notebooks which is essential for reproducibility.
- Model-generated examples provide detailed insights into what the prompts and generated outputs were.


Weakness:

- One might argue that technical novelty is limited in the paper with not much significant research contribution.
- The vast claims of zero-shot applications go on for 4 pages (Page 6-9) without much experiments and analysis to support them.
- Continuing on the point above, I believe that the paper needs a major revision where Section 4 could be expanded in more detail with more experiments. For example, Appendix B.2 provides nice examples of ablation where information from VLM is left out and should be a part of the main section compared to unjustified claims in Section 5. Even some applications from Section 5 could be absorbed in Section 4 only when provided with supporting experiments and ablation studies.
- This paper seems like a mix of position and experimental paper while doing justice to neither.
- Even though the zero-shot results might be a promising direction, currently the 0-shot results seem far from industry standard applications where there is a huge gap between all the automatic metrics from the baseline method (6.9 vs 40.97 BLEU, 44.5 vs 152 CIDEr)
- It is not clear how prompts were designed for few shot methods like the ones described in Section 4.1
- Similar experiments with few-shot techniques for Section 4.2 and 4.3 would have made the results more compelling.
- Even with few-shot learning, have the authors experimented with providing more than 3 examples and comparing the evaluation results as the number of examples is increased.
- Table 1, Bleu-4 of exact 0 value for ZeroCap model seems like a bug in the experiments. Could the authors please recheck?
- It is not clear how the videos are processed in Section 4.3? Does the model work on a per frame basis? How is CLIP(caption) computed and how does it differ from CLIP(video)?
- Human evaluation is not provided for the different experiments.


Questions:
- Could the authors discuss and clarify how they design these multimodal prompts?
- As the authors also discuss in Section 2 (page 3), simple prompts modifications have a huge impact on performance. Did the authors thus experiment with different wording of the prompts?
- An ablation on different prompts would be more compelling. How would the results compare when object, place categories or the number of people is omitted from the prompt in Section 4.1? Does the model always generate 3 object and place categories from the VLM?
- In Section 4.1 (Page 4), have the authors also experimented with nucleus sampling or beam search? How do the results compare against the greedy search?
- Have the authors also experimented with different prompts for description generation against captioning bot? Example prompting with `I am an intelligent image describing bot` compared to `image captioning bot` to promote longer descriptions and differentiate between the two tasks?
- Could the authors clarify how the long-transcripts subset of the dataset was created and the statistics of the dataset?


Suggestions/Comments:
- Please take care of using citet compared to citep and vice versa (natbib style) appropriately (especially in Section 7 and Appendix).
- Section 1 can be improved by providing relevant citations on page 2 of the baselines producing results 11.3 and 40.7 R@1 even though they are discussed much later in Section 4.2 on Page 5.
- Pictorial depiction of the models either in the main section or in the appendices would increase the understanding and interpretability of the proposed approach.


**Summary Of The Paper:**

This paper explores multimodal prompt engineering for zero-shot multimodal reasoning. Using language as the intermediate representation, the authors propose Socratic Models (SM) where outputs from different modalities are composed into textual prompts to provide zero-shot results on image captioning, description generation and video-to-text retrieval. The paper also discusses potential applications of these methods for dialog, QA and robot perception and planning.

**Summary Of The Review:**

The results seem promising; however, the paper could be improved in the presentation and more detailed descriptions. Please see strengths, weaknesses, suggestions and comments. It would be nice to see a revised version of the paper in the future conferences.

--- Rebuttal update ---

I believe the authors have invested fair amount of time to address most of the questions/weaknesses. Even though the 0-shot results are pretty low, they still serve as baseline against other related work. Based on the rebuttal, I am increasing my score.

---

> ### Author Response · Authors · 2022-11-17
> **Response to Reviewer J1Ki (1/3)**
>
> We thank the reviewer for their feedback, and for taking the time to evaluate our manuscript. As suggested by the reviewer, we are expanding Section 4 with more quantitative experiments and ablations.  We hope that these, as well as answers to questions below, can address some of the questions and suggestions offered by the reviewer.
>
> We wanted to first post these here to be timely and allow some time for discussion if desired – we’re working on adding them to the manuscript .pdf.
>
> > *"I believe that the paper needs a major revision where Section 4 could be expanded in more detail with more experiments. For example, Appendix B.2 provides nice examples of ablation where information from VLM is left out and should be a part of the main section…"*
>
> Thank you for the suggestion. Along these lines, we have included more experiments in this response which expand on Section 4.  This includes several suggestions from other comments, including:
> * Trying different wordings of prompts.
> * Ablating the prompts by removing subsets of multimodal information, including getting rid of the "object"/"place"/ etc. categories.
> * Experimenting with different sampling (with temperature vs. greedy vs. nucleus).
> * Increasing the # for "few-shot" to more than 3.
> * Trying few-shot experiments on an additional section (4.2).
> * Bringing up the experiments of Appendix B.2 into the main paper.
>
> The details of these additional experiments in Section 4 are described in comments below.  Also, please note that overall, our baseline numbers have slightly improved since submission (due to improvements to the OpenAI LLM model behind the same APIs), so in order to run these comparisons we recomputed the baseline numbers.
>
> Also, in domains that are data rich (as in Section 4), we also agree that *"currently the 0-shot results seem far from industry standard"* and for the moment, they can only serve as baselines. We accordingly shaped our experiments to primarily focus on comparisons to other published zero-shot methods.
>
> > *"Could the authors discuss and clarify how they design these multimodal prompts?... Did the authors thus experiment with different wording of the prompts?... An ablation on different prompts would be more compelling… How would the results compare when object, place categories or the number of people is omitted…"*
>
> As requested, we added a few more ablations on the prompts accordingly (results below). We followed common practices among prompt engineering for reasoning **[1][2]** to find text templates suitable for in-context completion. In particular, "I am an intelligent image captioning bot" was specifically inspired by the prompt "intelligent question answering bot" in distribution of the data used to finetune InstructGPT for open QA **[3]**.
>
> We provide experiments both with (i) changing the wording of the prompt, and (ii) ablating the different object and place categories – we think these are both great suggestions.
>
>
> | Prompt Changes                | BLEU-4 | METEOR | CIDEr | SPICE | ROUGE-L |
> | ----------------------------- | ------ | ------ | ----- | ----- | ------- |
> | Ours (original)               | 9.95   | 16.17  | 50.05 | 10.82 | 36.13   |
> | *Wording changes*
> | remove "likely, short"        | 5.45   | 16.72  | 35.32 | 11.02 | 31.33   |
> | replace "short" w/ "creative" | 5.49   | 16.29  | 32.28 | 10.23 | 30.22   |
> | remove "intelligent"          | 10.70  | 16.61  | 56.39 | 10.52 | 36.54   |
> | *Categories ablations*
> | remove object categories      | 4.33   | 11.32  | 25.86 | 6.54  | 28.98   |
> | remove place categories       | 7.99   | 16.77  | 47.05 | 10.27 | 36.32   |
> | remove number of people       | 8.93   | 17.23  | 48.69 | 11.56 | 36.63   |
> |
>
> For (i), changing the prompt from "a likely, short caption" to "a caption" or to "a likely, creative caption" decreases performance. We also observe that the captions generated by these alternatives tend to be overly verbose, and are less likely to match the distribution of (short) captions that are labeled in the COCO dataset. Surprisingly, removing "intelligent" from the description "intelligent image captioning bot" seems to slightly improve performance on this dataset.
>
> For (ii), removing entities from the VLM (objects, places, # people) also tends to reduce performance, most substantially when object categories are removed.  Removing the number of people performs more comparably against the original prompt.
>
> ---
>
> **[1]** *Takeshi Kojima et al. "Large Language Models are Zero-Shot Reasoners" NeurIPS, 2022.*
>
> **[2]** *Jiachang Liu et al. "What Makes Good In-Context Examples for GPT-3?" DeeLIO, 2022.*
>
> **[3]** *Long Ouyang et al. "Training language models to follow instructions with human feedback." NeurIPS, 2022.*

---

> > ### Author Response · Authors · 2022-11-17
> > **Response to Reviewer J1Ki (2/3)**
> >
> > > *"In Section 4.1 (Page 4), have the authors also experimented with nucleus sampling or beam search? How do the results compare against the greedy search?"*
> >
> > Here are additional ablations compared to greedy search, and nucleus sampling (which performs favorably over beam search for captioning tasks according to Junnan Li et al. 2022 **[4]**). We observe that sampling with temperature performs comparably (slightly better for most metrics) versus both alternatives.
> >
> > | Sampling Method                      | BLEU-4 | METEOR | CIDEr | SPICE | ROUGE-L |
> > | ------------------------------------ | ------ | ------ | ----- | ----- | ------- |
> > | N=20, Temp = 0.9 Sampling (original) | **9.95**   | 16.17  | **50.05** | **10.82** | 36.13   |
> > | N=20, Nucleus Sampling (top_p = 0.9) | 8.49   | **16.64**  | 47.69 | 10.75 | 36.44   |
> > | N=1, Greedy Search                   | 9.33   | 15.90  | 49.53 | 10.46 | **36.88**   |
> > |
> >
> >
> > > *"It is not clear how prompts were designed for few shot methods like the ones described in Section 4.1"*
> >
> > Agreed this was not clearly specified, here is how it’s done and we will clarify in the manuscript.  There is no additional prompt design compared to zero-shot, it’s just that a few ground-truth examples are added to the prompt, as is done for few-shot language model examples.  Here’s a comparison of a zero-shot and 1-shot prompt to illustrate:
> >
> > Zero-shot prompt:
> > ```
> > I am an intelligent image captioning bot. This image is a {img_type}. There {num_people}... <trimmed for brevity>  …a  caption I can generate to describe this image is:
> > ```
> >
> > 1-shot prompt:
> > ```
> > I am an intelligent image captioning bot. This image is a photo. There are 3 people… <trimmed for brevity> …a  caption I can generate to describe this image is: Three friends sitting on the beach together.
> >
> > I am an intelligent image captioning bot. This image is a {img_type}. There {num_people}... <trimmed for brevity>  …a  caption I can generate to describe this image is:
> > ```
> >
> >
> > > *"have the authors experimented with providing more than 3 examples"*
> >
> > We’ve run these experiments and find that captioning performance tends to asymptote with more than 3 random examples (we tried 8 as well, results shown below), though future work may investigate dynamic few-shot prompts to retrieve few-shot examples that maximize test time performance.
> >
> > | Method | BLEU-4 | METEOR | CIDEr | SPICE | ROUGE-L |
> > | ------ | ------ | ------ | ----- | ----- | ------- |
> > | 0-shot | 9.95   | 16.17  | 50.05 | 10.82 | 36.13   |
> > | 3-shot | **18.21**  | **20.49**  | **76.26** | 13.92 | **43.72**   |
> > | 8-shot | 15.44  | 20.39  | 73.02 | **14.88** | 43.09   |
> > |
> >
> > ---
> >
> > **[4]**  *Junnan Li et al. "BLIP: Bootstrapping Language-Image Pre-training for Unified Vision-Language Understanding and Generation." ICML, 2022.*

---

> > > ### Author Response · Authors · 2022-11-17
> > > **Response to Reviewer J1Ki (3/3)**
> > >
> > > > *"Similar experiments with few-shot techniques for Section 4.2 and 4.3 would have made the results more compelling."*
> > >
> > > This is a good suggestion, we’ve run few-shot examples for Section 4.2 on Concadia (contextual captioning benchmark), here are the results (CIDEr scores):
> > >
> > > | Method | Captions | Descriptions |
> > > | ------ | -------- | ------------ |
> > > | 0-shot |   38.8   |     23.0     |
> > > | 3-shot |   59.6   |     27.3     |
> > > |
> > >
> > > > (Re: Section 4.2) *"Have the authors also experimented with different prompts for description generation against captioning bot? Example prompting with I am an intelligent image describing bot compared to image captioning bot to promote longer descriptions and differentiate between the two tasks?"*
> > >
> > > Replacing the prompt from `intelligent image captioning bot` to `intelligent image describing bot` appears to slightly reduce performance (results below). Interestingly, we observe that the generated descriptions are quite similar to those generated with "captioning" in the prompt – and while more verbose in some cases, do not necessarily add new information on visual details. It may be interesting future work to explore additional ways to extract perceptual cues from the VLM.
> > >
> > > | Prompt Changes                    | CIDEr |
> > > | --------------------------------- | ----- |
> > > | `image captioning bot` (original) | 23.0  |
> > > | `image describing bot`            | 21.3  |
> > > |
> > >
> > > > *"Table 1, Bleu-4 of exact 0 value for ZeroCap model seems like a bug in the experiments."*
> > >
> > > On this n=100 test set, we checked ZeroCap's BLEU-4 score to be 3.79e-06 (we rechecked and re-ran evaluation to confirm this was correct). Relatively, this is not too far from a BLEU-4 score of 2.6, which matches what was reported in ZeroCap's original paper.
> > >
> > >
> > > > *"It is not clear how the videos are processed in Section 4.3?"*
> > >
> > > The videos are processed in the same way as in Portillo-Quintero et al., 2021, which averages the CLIP image features of every frame – we refer to this as CLIP(video). CLIP(caption) is the CLIP text features of the caption. We are working on adding a few notes in the manuscript to help clarify these points.
> > >
> > >
> > > > *"Could the authors clarify how the long-transcripts subset of the dataset was created and the statistics of the dataset?"*
> > >
> > > The long-transcripts subset consists of 1007 videos with transcript length (> 145 characters). The average character length of these transcripts are ~245 ± 87 characters (our anonymous colabs allow anyone to compute additional statistics on this subset in case needed).
> > >
> > >
> > > > *"Please take care of using citet compared to citep and vice versa (natbib style) appropriately (especially in Section 7 and Appendix)."*
> > >
> > > Thanks for pointing this out, we have corrected this accordingly.
> > >
> > >
> > > > *"Section 1 can be improved by providing relevant citations on page 2 of the baselines producing results 11.3 and 40.7 R@1 even though they are discussed much later in Section 4.2 on Page 5."*
> > >
> > > Thanks! We have added these suggested citations.
> > >
> > >
> > > > *"Pictorial depiction of the models either in the main section or in the appendices would increase the understanding and interpretability of the proposed approach."*
> > >
> > > Thank you, we are working on adding pictorial depictions for each into the appendix.

---

> > > > ### Comment · Reviewer_J1Ki · 2022-12-02
> > > > **Rebuttal update**
> > > >
> > > > Thank you authors for the detailed response and experiments. Good job!
> > > > I have read through your responses and I feel most of my comments have been incorporated and addressed.
> > > > With these explanations in the main paper and based on the feedback from the fellow reviewers, I am also leaning towards increasing my score and acceptance of the paper.
> > > >
> > > > Additionally, I want to leave one more suggestion to provide more details about Portillo-Quintero et al., 2021 (eg. describing the sampling of frames in the video and how the model is trained and evaluated in run-time). This could help the readers working in the domain of video processing for real-life applications.

---

### Official Review · Reviewer_3Nxe · 2022-10-27

**Confidence:** 4
**Correctness:** 3
**Technical Novelty And Significance:** 3
**Empirical Novelty And Significance:** 3
**Recommendation:** 8

**Clarity, Quality, Novelty And Reproducibility:**

The paper is clearly written and easy to follow. Authors also provided links to Colab where their work could be easily reproduced.

**Strength And Weaknesses:**

Pros:

The paper is generally easy to follow, well motivated, and shows creative tricks to prompt combination of multi-modal models and LLM to solve multi-modal reasoning tasks. The paper attempted to formulate the multi-modal prompt engineering as traversal of a computational graph which is novel and intuitive. The paper also shows good zero-shot performance over strong baseline.

Cons:
While the proposed multi-modal prompting is useful to prompt constrained LLM, VLM, ALM, figuring out the model sequence and the prompt structure to get the best outcome is not trivial. For the cases where the training data is publicly available, could it be possible wouldn't it be best to to just training task specific models. Also, in the case where the multi-modal data is not possible to generate, e.g, egocentric perception, multi-modal assistive dialog, and robotic perception and planning, could it be more useful to use this technique to generate a multi-modal training data that can be used to train task specific model?

**Summary Of The Paper:**

This paper expands prompt engineering techniques to multi-modality scenarios. Specifically, the paper sought to find how best to prompt a set of unimodal and multi-modal models to solve multi-modal tasks that are difficult to solve independently or would require large amount of training data to tackle. The paper also creatively demonstrated the power of in-context substitution where information from non-language domain is substituted into an LLM for contextual reasoning. The paper applied the developed techniques to different multi-modal tasks including image captioning, contextual image description, video-to-text retrieval, and others such as egocentric perception, multi-modal assistive dialog, and robotic perception and planning. The paper also introduce a novel zero-shot evaluation techniques that could be adopted for model selection purposes.

**Summary Of The Review:**

The paper demonstrated that multi-modal prompt engineering is a viable option for utilizing publicly available multi-modal models at zero-shot without retraining a new model for a specific multi-modal reasoning task. The paper is well motivated and experimental results looks convincing.

---

> ### Author Response · Authors · 2022-11-17
> **Response to Reviewer 3Nxe**
>
> We thank the reviewer for their encouraging comments, including their opinion that the paper *"creatively demonstrated the power of in-context substitution"*.  We've made some valuable additions to our work based on all the reviewers' feedback, as well as a few responses below to some of the comments from the reviewer.
>
> > *"The paper is generally easy to follow, well motivated, and shows creative tricks to prompt combination of multi-modal models and LLM to solve multi-modal reasoning tasks. The paper attempted to formulate the multi-modal prompt engineering as traversal of a computational graph which is novel and intuitive."*
>
> Thank you!
>
> > *"While the proposed multi-modal prompting is useful to prompt constrained LLM, VLM, ALM, figuring out the model sequence and the prompt structure to get the best outcome is not trivial."*
>
> The reviewer may be interested in our responses to Reviewer J1Ki, where we added additional experiments per their requests to try different types of prompt wording / other attributes ablations.
>
> > *"For the cases where the training data is publicly available, could it be possible wouldn't it be best to to just training task specific models."*
>
> We agree that for certain applications, doing large-scale training of joint multimodal models may be preferred – but those regimes may need to have all of: (i) abundant training data available for the multimodal task of interest, (ii) full access to model code for the component foundation models in order to generally wire up new models and do finetuning with gradient-based methods, and (iii) lots of compute available. It’s likely that if one or more of these (i, ii, iii) conditions aren't met, then doing an approach like demonstrated in our paper may be a reasonable alternative.
>
> > *"...could it be more useful to use this technique to generate a multi-modal training data that can be used to train task specific model?"*
>
> It’s an interesting idea for future work, and may also for example be interesting from the inference-speed perspective, to be able to “distill” a large multi-modal model into a fast, smaller multimodal model.

---

### Decision · Program_Chairs · 2023-01-20

**Decision:**

Accept: notable-top-25%

**Justification For Why Not Higher Score:**

I'm not fully convinced that this merits an oral, with the caveat that it depends on the quality of the other papers.

**Justification For Why Not Lower Score:**

This could be a poster, but I feel it should be accepted.

**Metareview: Summary, Strengths And Weaknesses:**

The paper investigates techniques for prompting pretrained unimodal and multimodal models to perform multimodal tasks in a zero-shot fashion. The paper proposes Socratic Models as a modular architecture that composes the outputs of various pretrained models into textual prompts for large multimodal models. The paper explores the use of Socratic Models in the context of various multimodal tasks, including image captioning, video-to-text retrieval, egocentric (robot) perception, and multimodal dialogue, with zero-shot performance that exceeds that of several baselines and is comparable to some one-shot baselines.

The paper was reviewed by referees who agree that the paper explores an interesting and important problem (i.e., zero-shot mulitimodal reasoning). As they emphasize, the proposed approaches for combining the output of pretrained models as a means of prompting a large mulitimodal model are both creative and effective, as evidenced by detailed insights into the prompts and outputs for the various tasks considered.  One of the core contribution of the paper is the demonstration that multimodal prompt engineering is viable using publicly available multimodel models in a zero-shot manner. Another is the establishment of solid baselines that would be useable by future researchers, which the authors support by providing publicly available implementations. The reviewers identified a few issues with the original submission, notably the concern that identifying the model sequence and prompt structure suitable to a given problem is non-trivial, as well as questions about the zero-shot performance and the need for additional few-shot baselines. The authors made a considerable effort to address these questions/concerns, which the reviewers acknowledged. Overall, this is solid work that provides a valuable contribution to the community.

**Note From Pc:**

if the above contains the word "oral" or "spotlight" please see: "oral" presentation means -> notable-top-5% and "spotlight" means -> notable-top-25%. As stated in our emails, we are disassociating presentation type from AC recommendations

**Summary Of Ac-Reviewer Meeting:**

N/A